# On Kernelized Multi-Armed Bandits with Constraints

**Xingyu Zhou**
Electrical and Computer Engineering
Wayne State University
Detroit, MI, USA
xingyu.zhou@wayne.edu

**Bo Ji**
Computer Science
Virginia Tech
Blacksburg, VA, USA
boji@vt.edu

## Abstract

We study a stochastic bandit problem with a general unknown reward function and a general unknown constraint function. Both functions can be non-linear (even non-convex) and are assumed to lie in a reproducing kernel Hilbert space (RKHS) with a bounded norm. In contrast to safety-type hard constraints studied in prior works, we consider soft constraints that may be violated in any round as long as the cumulative violations are small. Our ultimate goal is to study how to utilize the nature of soft constraints to attain a finer complexity-regret-constraint trade-off in the kernelized bandit setting. To this end, leveraging primal-dual optimization, we propose a general framework for both algorithm design and performance analysis. This framework builds upon a novel sufficient condition, which not only is satisfied under general exploration strategies, including *upper confidence bound* (UCB), *Thompson sampling* (TS), and new ones based on *random exploration*, but also enables a unified analysis for showing both sublinear regret and sublinear or even zero constraint violation. We demonstrate the superior performance of our proposed algorithms via numerical experiments based on both synthetic and real-world datasets. Along the way, we also make the first detailed comparison between two popular methods for analyzing constrained bandits and Markov decision processes (MDPs) by discussing the key difference and some subtleties in the analysis, which could be of independent interest to the communities.

## 1 Introduction

Stochastic bandit optimization of an unknown function $f$ has recently gained increasing popularity due to its widespread real-life applications such as recommendations [1], cloud resource configurations [2], and wireless power control [3]. At each time $t$, an action $x_t$ is selected and then a (noisy) bandit reward feedback $y_t$ is observed. The goal is to maximize the cumulative reward, or equivalently minimize the total regret due to not choosing the optimal action in hindsight. To capture a generic function, researchers have turned to nonparametric models of $f$ via Gaussian process or reproducing kernel Hilbert space (RKHS), which are able to uniformly approximate an arbitrary continuous function over a compact set [4]. In this paper, as in [5, 6], we consider the agnostic setting (i.e., frequentist-type), where $f$ is assumed to be a fixed function in an RKHS with a bounded norm (i.e., a measure of smoothness). We call this setting *frequentist-type kernelized bandits* (KB).

In addition to a generic non-linear or even non-convex function $f$, another common feature in practical applications is that there often exist additional constraints in the decision-making process such as hard-constraint like safety or soft-constraint like cost. To this end, there have been exciting recent advances in the theoretical analysis of constrained kernelized bandits. In particular, [7–9] propose algorithms with convergence guarantees, while [10], to the best our knowledge, is the first

work that establishes regret bounds for their developed algorithm, although under the Bayesian-type[1] setting. These algorithms mainly focus on KB with a *hard* constraint such as safety, i.e., the selected action in *each round* needs to satisfy the constraint with a high probability. Hence, compared to the unconstrained case, additional computation is required to construct a *safe* action set in each round, which not only incurs additional complexity burdens, but often leads to conservative performance.

**Motivations.** In practice, there are also many applications that involve *soft* constraints that may be violated in any round. The goal is to maximize the total reward while minimizing the total constraint violations. For example, in a wireless networking system, the reward could be the throughput while the constraint is that the average energy consumption is below a threshold. In this case, the constraint can be violated at any time, and moreover, using less energy at one time instant allows one to use more in the next time instant, i.e., "compensation" across time (see a formal definition in Eq.(1)) Furthermore, existing provably efficient algorithms [7–10] largely build on upper confidence bound (UCB) exploration, which often has inferior empirical performance compared to Thompson sampling (TS) exploration. Hence, another key question is whether one can design provably efficient CKB algorithms with general explorations. In summary, the following fundamental theoretical question remains open:

*Can a finer complexity-regret-constraint trade-off be attained in CKB under general explorations?*

**Contributions.** In this paper, we take a systematic approach to affirmatively answer the above fundamental question. In particular, we tackle the complexity-regret-constraint trade-off by formulating KB under soft constraints as a stochastic bandit problem where the objective is to maximize the cumulative reward while minimizing the cumulative constraint violations and maintaining the same computation complexity as in the unconstrained case. Our detailed contributions are as follows.

- We develop a unified framework for CKB based on primal-dual optimization, which can guarantee both sublinear reward regret and sublinear total constraint violation under a class of general exploration strategies, including UCB, TS, and new effective ones (e.g., random exploration) under the same complexity as the unconstrained case. We also show that by introducing slackness in the dual update, one can trade regret to achieve bounded or even zero constraint violation. This framework builds upon a novel sufficient condition, which not only facilitates the design of new CKB algorithms but provides a unified view in the performance analysis.

- We demonstrate the superior performance of our proposed algorithms via numerical experiments based on both synthetic and real-world data. In addition, we discuss the benefits of our algorithms in terms of various practical considerations such as low complexity, scalability, robustness, and flexibility.

- We also provide the first detailed comparison between two popular methods for analyzing constrained bandits in general. Specifically, the first one is based on convex optimization tool as in [11, 12], which is also the inspiration for our paper. The other one is based on Lyapunov-drift argument as in [13–15]. We discuss the key difference in terms of regret and constraint violation analysis in these two methods and highlight the subtlety in applying a standard queueing technique (i.e., Hajek lemma [16]) to bound the constraint violation in the second method. We believe this provides a clear picture on the methodology, which is of independent interest to the communities.

## 1.1 Related Work

In the special cases of KB, including multi-armed bandits (MAB) and linear bandits (e.g., KB with a linear kernel), there is a large body of work on bandits with different types of constraints, including knapsack bandits [17–19], conservative bandits [20–22], bandits with fairness constraints [23, 24], bandits with hard safety constraints [25–27], and bandits with cumulative soft constraints [28, 13]. Among them, the bandit setting with cumulative soft constraints is the closest to ours. In particular, [13] considers linear bandits under UCB exploration and a zero constraint violation is attained via the Lyapunov-drift method. However, it is unclear how to generalize it to handle general exploration strategies such as TS; see a further discussion in Section 4.

---

[1] In the Bayesian-type KB, $f$ is assumed to be a sample from a Gaussian process and the observation noise is Gaussian. In contrast, in our considered frequentist-type KB, $f$ is a fixed function in an RKHS and the noise can be any sub-Gaussian. A better regret bound can often be achieved in the easier Bayesian-type KB setting [6].

Broadly speaking, our work is also related to reinforcement learning (RL) with soft constraints, i.e., constrained MDPs. In particular, our analysis is inspired by those on constrained MDPs [11, 12] (which is another popular method to handle constrained bandits and MDPs via convex optimization tools), but has significant differences. First, in those works, the constraint violation is $\widetilde{O}(\sqrt{T})$. In contrast, ours can attain bounded and even zero constraint violations by introducing the slackness in the dual update. Second, they only consider UCB-type exploration, but our algorithms can be equipped with various exploration strategies (including UCB), thanks to our general sufficient condition. Third, they focus on either tabular or linear function approximation settings. In contrast, both objective and constraint functions we consider can be *nonlinear*. There are also recent works on constrained MDPs that claim to achieve bounded or zero constraint violation [14, 15] based on the Lyapunov-drift method. However, as in the bandit case, it is unclear how to generalize it to handle general explorations beyond UCB. Finally, the theoretical understanding of unconstrained KB (a.k.a. Gaussian process bandit) is well-established, including regret upper bounds under UCB and TS [5, 29, 6] and lower bounds [30, 31]. We remark that our work is mainly a theory-guided study. In a more practical area of KB, i.e., Bayesian optimization (BO), there have been many BO algorithms developed for the constrained setting; see [32–34] and the references therein. Although these algorithms enjoy good performance in various practical settings, their theoretical performance guarantees are still unclear.

## 2  Problem Formulation and Preliminaries

We consider a stochastic bandit optimization problem with *soft* constraints. In particular, in each round $t \in \{1, 2, \ldots, T\}$, a learning agent chooses an action $x_t \in \mathcal{X}$ and receives a bandit reward feedback $r_t = f(x_t) + \eta_t$, where $\eta_t$ is a zero-mean noise. The learning agent also observes a bandit constraint feedback $c_t = g(x_t) + \xi_t$, where $\xi_t$ is a zero-mean noise. By soft constraints, the goal here is to maximize the cumulative reward while minimizing the cumulative total violation.

**Learning Problem.** Define cumulative regret and constraint violation as follows:

$$\mathcal{R}(T) := Tf(x^*) - \sum_{t=1}^{T} f(x_t), \quad \mathcal{V}(T) := \left[ \sum_{t=1}^{T} g(x_t) \right]_+, \tag{1}$$

where $x^* := \operatorname{argmax}_{\{x \in \mathcal{X}: g(x) \leq 0\}} f(x)$ and $[\cdot]_+ := \max\{\cdot, 0\}$. The goal of the learning agent is to achieve both sublinear regret and sublinear constraint violation. In fact, we will establish bounds on the following stronger version of regret. Specifically, let $\pi$ be a probability distribution over the set of actions $\mathcal{X}$ (i.e., stochastic policy), and let $\mathbb{E}_\pi[f(x)] := \int_{x \in \mathcal{X}} f(x)\pi(x)\,dx$ and $\mathbb{E}_\pi[g(x)] := \int_{x \in \mathcal{X}} g(x)\pi(x)\,dx$. We compare our achieved reward with the following baseline optimization problem: $\max_\pi \{\mathbb{E}_\pi[f(x)] : \mathbb{E}_\pi[g(x)] \leq 0\}$ where both $f$ and $g$ are known, and $\pi^*$ is its optimal solution. Now, our stronger regret is defined as

$$\mathcal{R}_+(T) := T\mathbb{E}_{\pi^*}[f(x)] - \sum_{t=1}^{T} f(x_t). \tag{2}$$

Clearly, we have $\mathcal{R}(T) \leq \mathcal{R}_+(T)^2$. Throughout the paper, we assume the following commonly used condition in constrained optimization literature; see also [13, 35, 11].

**Assumption 1** (Slater's condition). *There is a constant $\delta > 0$ such that there exists a probability distribution $\pi_0$ that satisfies $\mathbb{E}_{\pi_0}[g(x)] \leq -\delta$. Without loss of generality, we assume $\delta \leq 1$.*

This is a quite mild assumption since it only requires that one can find a stochastic policy under which the expected cost is less than a strictly negative value. This is in sharp constraint to existing KB algorithms for hard constraints that typically require the existence of an initial safe action [9, 10].

In this paper, we consider the frequentist-type regularity assumption that is typically used in uncontrained KB works (e.g., [5, 6]). Specifically, we assume that $f$ is a fixed function in an RKHS with a bounded norm. In particular, the RKHS for $f$ is denoted by $\mathcal{H}_k$, which is completely determined by the corresponding kernel function $k : \mathcal{X} \times \mathcal{X} \to \mathbb{R}$. Any function $h \in \mathcal{H}_k$ satisfies the *reproducing property*: $h(x) = \langle h, k(\cdot, x) \rangle_{\mathcal{H}_k}$, where $\langle \cdot, \cdot \rangle_{\mathcal{H}_k}$ is the inner product defined on $\mathcal{H}_k$. Similarly, for the unknown constraint function $g$, we assume that $g$ is a fixed function in the RKHS defined by a

---

[2]This stronger regret also allows us to use convex optimization tool since $\mathbb{E}_\pi[f(x)] := \int_{x \in \mathcal{X}} f(x)\pi(x)\,dx$ is a convex function with respect to $\pi$, which will be important for our analysis.

kernel function $\widetilde{k}$, and the RKHS for $g$ is denoted by $\mathcal{H}_{\widetilde{k}}$. We assume that the following boundedness property holds throughout the paper.

**Assumption 2** (Boundedness). *We assume that $\|f\|_{\mathcal{H}_k} \leq B$ and $k(x, x) \leq 1$ for any $x \in \mathcal{X}$ and that the noise $\eta_t$ is i.i.d. $R$-sub-Gaussian. Similarly, we assume that $\|g\|_{\mathcal{H}_{\widetilde{k}}} \leq G$ and $\widetilde{k}(x, x) \leq 1$ for any $x \in \mathcal{X}$ and that the noise $\xi_t$ is i.i.d. $\widetilde{R}$-sub-Gaussian.*

**Gaussian Process Surrogate Model.** We use a Gaussian process (GP), denoted by $\mathcal{GP}(0, k(\cdot, \cdot))$, as a prior for the unknown function $f$, and a Gaussian likelihood model for the noise variables $\eta_t$, which are drawn from $\mathcal{N}(0, \lambda)$ and are independent across $t$. Note that this GP surrogate model is used for algorithm design only; it does not change the fact that $f$ is a fixed function in $\mathcal{H}_k$ and that the noise $\eta_t$ can be sub-Gaussian (i.e., an *agnostic* setting [6]). Let $[t] := \{1, 2, \ldots, t\}$. Conditioned on a set of observations $H_t = \{(x_s, r_s), s \in [t]\}$, by the properties of GP [36], the posterior distribution for $f$ is $\mathcal{GP}(\mu_t(\cdot), k_t(\cdot, \cdot))$, where

$$\mu_t(x) := k_t(x)^T (K_t + \lambda I)^{-1} R_t, \tag{3}$$

$$k_t(x, x') := k(x, x') - k_t(x)^T (K_t + \lambda I)^{-1} k_t(x'), \tag{4}$$

in which $k_t(x) := [k(x_1, x), \ldots, k(x_t, x)]^T$, $K_t := [k(x_u, x_v)]_{u,v \in [t]}$, and $R_t$ is the (noisy) reward vector $[r_1, r_2, \ldots, r_t]^T$. In particular, we also define $\sigma_t^2(x) := k_t(x, x)$. Let $K_A := [k(x, x')]_{x,x' \in A}$ for $A \subseteq \mathcal{X}$. We define the maximum information gain as $\gamma_t(k, \mathcal{X}) := \max_{A \subseteq \mathcal{X}:|A|=t} \frac{1}{2} \ln |I_t + \lambda^{-1} K_A|$ where $I_t$ is the $t \times t$ identity matrix. The maximum information gain plays a key role in the regret bounds of GP-based algorithms. While $\gamma_t(k, \mathcal{X})$ depends on the kernel $k$ and domain $\mathcal{X}$, we simply use $\gamma_t$ whenever the context is clear. For instance, if $\mathcal{X}$ is compact and convex with dimension $d$, then we have $\gamma_t = O((\ln t)^{d+1})$ for squared exponential kernel $k_{\text{SE}}$, $\gamma_t = O(d \ln t)$ for linear kernel [6] and $\gamma_t = \widetilde{O}(T^{\frac{d}{2\nu+d}})$ (where $\nu$ is a hyperparameter) for Matérn kernel $k_{\text{Matérn}}$ [37]. Similarly, the learning agent also uses a GP surrogate model for $g$, i.e., a GP prior $\mathcal{GP}(0, \widetilde{k}(\cdot, \cdot))$ and a Gaussian noise $\mathcal{N}(0, \widetilde{\lambda})$. Conditioned on a set of observations $\widetilde{H}_t = \{(x_s, c_s), s \in [t]\}$, the posterior distribution for $g$ is $\mathcal{GP}(\widetilde{\mu}_t(\cdot), \widetilde{k}_t(\cdot, \cdot))$, where $\widetilde{\mu}_t$ and $\widetilde{k}_t$ are computed in the same way.

## 3 A Unified Framework for Constrained Kernelized Bandits

In this section, leveraging primal-dual optimization, we develop a unified framework for both algorithm design and performance analysis in constrained kernelized bandits. In particular, we first propose a "master" algorithm called CKB (constrained KB), which can be equipped with very general exploration strategies. Then, we develop a novel sufficient condition, which not only provides a unified analysis of regret and constraint violation, but also facilitates the design of new exploration strategies (and hence new CKB algorithms) with rigorous performance guarantees.

**Algorithm.** We first explain our "master" algorithm CKB in Algorithm 1, which is based on primal-dual optimization. Let the Lagrangian of the baseline problem $\max_\pi \{\mathbb{E}_\pi[f(x)] : \mathbb{E}_\pi[g(x)] \leq 0\}$ be $\mathcal{L}(\pi, \phi) := \mathbb{E}_\pi[f(x)] - \phi \mathbb{E}_\pi[g(x)]$ and the associated dual problem is defined as $\mathcal{D}(\phi) := \max_\pi \mathcal{L}(\pi, \phi)$ with the optimal dual variable being $\phi^* := \text{argmin}_{\phi \geq 0} \mathcal{D}(\phi)$. Note that since both $f$ and $g$ are unknown, the agent has to first generate estimates of them (i.e., $f_t$ and $g_t$, respectively) based on exploration strategies $\mathcal{A}_f$ and $\mathcal{A}_g$, which capture the tradeoff between exploration and exploitation (line 3). Then, both estimates will be truncated according to the range of $f$ and $g$, respectively (lines 4-5) (where Proj is the projection operator). The truncation is necessary for our analysis, but it does not impact the regret bound since it will not lead to loss of useful information. Then, lines 6-7 correspond to the primal optimization step that approximates $\mathcal{D}(\phi_t)$ (i.e., approximate $\mathcal{L}$ by $\bar{\mathcal{L}}$ with $f$ and $g$ replaced by $\bar{f}_t$ and $\bar{g}_t$). The reason behind line 7 is that one of the optimal solutions for $\max_\pi \bar{\mathcal{L}}(\pi, \phi_t)$ is simply $\text{argmax}_x(\bar{f}_t(x) - \phi_t \bar{g}_t(x))$. Then, line 8 is the dual update that minimizes $\mathcal{D}(\phi_t)$ with respect to $\phi$ by taking a projected gradient step with $1/V$ being the step size. The parameter $\rho$ is chosen to be larger than the optimal dual variable $\phi^*$, and hence the projected interval $[0, \rho]$ includes the optimal dual variable. This can be achieved since the optimal dual variable is bounded under Slater's condition, and in particular, we have $\phi^* \leq (\mathbb{E}_{\pi^*}[f(x)] - \mathbb{E}_{\pi_0}[f(x)])/\delta$ by [38, Theorem 8.42]. Finally, line 8 is the posterior update via standard GP regression for both $f$ and $g$ as computed in (3) and (4) with $\sigma_t^2(x) = k_t(x, x)$ and $\widetilde{\sigma}_t^2(x) = \widetilde{k}_t(x, x)$.

**Remark 1** (Computational complexity). *CKB enjoys the same computational complexity as the standard unconstrained case (e.g., [5]) since the additional dual update is a simple projection and*

---

**Algorithm 1** CKB Algorithm

---

1: **Parameters:** $V$, $\rho$, $\phi_1 = 0$, $\mu_0(x) = \widetilde{\mu}_0(x) = 0$, $\sigma_0(x) = \widetilde{\sigma}_0(x) = 1, \forall x$, exploration strategies
   $\mathcal{A}_f$ and $\mathcal{A}_g$
2: **for** batch $t = 1, 2, \ldots$ **do**
3:      Based on posterior models, generate $f_t$ and $g_t$ using $\mathcal{A}_f$ and $\mathcal{A}_g$, respectively
4:      Truncate $f_t$ as $\bar{f}_t(x) = \text{Proj}_{[-B,B]} f_t(x)$
5:      Truncate $g_t$ as $\bar{g}_t(x) = \text{Proj}_{[-G,G]} g_t(x)$
6:      Pseudo-acquisition function: $\widehat{z}_{\phi_t}(x) = \bar{f}_t(x) - \phi_t \bar{g}_t(x)$
7:      Choose primal action $x_t = \text{argmax}_{x \in \mathcal{X}} \widehat{z}_{\phi_t}(x)$; observe $r_t$ and $c_t$
8:      Update dual variable: $\phi_{t+1} = \text{Proj}_{[0,\rho]} \left[ \phi_t + \frac{1}{V} \bar{g}_t(x_t) \right]$
9:      Posterior model: update $(\mu_t, \sigma_t)$ and $(\widetilde{\mu}_t, \widetilde{\sigma}_t)$ via GP regression using new data $(x_t, r_t, c_t)$
10: **end for**

---

*the primal optimization keeps the same flavor as the unconstrained case, i.e., without constructing a specific safe set as in existing constrained KB algorithms designed for hard constraints.*

We call CKB a "master" algorithm as it allows us to employ different exploration strategies (or called *acquisition functions*) (i.e., $\mathcal{A}_f$ and $\mathcal{A}_g$). Therefore, one fundamental question is: *How to design efficient exploration strategies such that favorable performance can be guaranteed?* In the following, we first use UCB-type exploration as an example to gain useful insights, which in turn will facilitate the development of a novel sufficient condition. This condition not only is satisfied under very general exploration strategies, but also enables a unified analytical framework for showing both sublinear regret and sublinear constraint violation in constrained kernelized bandits.

We first introduce standard UCB and TS explorations under GP as in [5].

**Definition 1** (GP-UCB and GP-TS Explorations)**.** *Suppose the posterior distribution for a black-box function $h$ in round $t$ is given by $\mathcal{GP}(\widehat{\mu}_{t-1}(\cdot), \widehat{k}_{t-1}(\cdot, \cdot))$ and $\widehat{\beta}_t$ is an increasing sequence.*

*(i) The estimate of $h$ in round $t$ under GP-UCB exploration is $h_t(\cdot) = \widehat{\mu}_{t-1}(\cdot) + \widehat{\beta}_t \widehat{\sigma}_{t-1}(\cdot)$, where $\widehat{\sigma}_{t-1}^2(x) := \widehat{k}_{t-1}(x, x)$ for all $x \in \mathcal{X}$.*

*(ii) The estimate of $h$ in round $t$ under GP-TS exploration is $h_t(\cdot) \sim \mathcal{GP}(\widehat{\mu}_{t-1}(\cdot), \widehat{\beta}_t^2 \widehat{k}_{t-1}(\cdot, \cdot))$.*

### 3.1 Warm Up: CKB with GP-UCB Exploration

In this section, we instantiate CKB with GP-UCB exploration called `CKB-UCB`, as a warm-up. In particular, in `CKB-UCB`, $\mathcal{A}_f$ is a GP-UCB exploration (see Definition 1) with a positive $\widehat{\beta}_t$ sequence (i.e., optimistic with respect to reward), and $\mathcal{A}_g$ is a GP-UCB exploration with a negative $\widehat{\beta}_t$ sequence (i.e., optimistic with respect to cost). This instantiation enjoys the following performance guarantee.

**Theorem 1.** *Suppose $\rho \geq 4B/\delta$, $V = G\sqrt{T}/\rho$, $\mathcal{A}_f$ is a GP-UCB exploration with $\widehat{\beta}_t = \beta_t := B + R\sqrt{2(\gamma_{t-1} + 1 + \ln(2/\alpha))}$, and $\mathcal{A}_g$ is a GP-UCB exploration with $\widehat{\beta}_t = -\widetilde{\beta}_t := -(G + R\sqrt{2(\widetilde{\gamma}_{t-1} + 1 + \ln(2/\alpha))})$. Under Slater's condition in Assumption 1 and regularity assumptions in Assumption 2, `CKB-UCB` achieves the following bounds simultaneously with probability at least $1 - \alpha$ for any $\alpha \in (0, 1)$:*

$$\mathcal{R}_+(T) = O\left( B\sqrt{T\gamma_T} + \sqrt{T\gamma_T(\gamma_T + \ln(2/\alpha))} + \rho G\sqrt{T} \right),$$

$$\mathcal{V}(T) = O\left( (1 + \frac{1}{\rho})\left( C\sqrt{T\widehat{\gamma}_T} + \sqrt{T\widehat{\gamma}_T(\widehat{\gamma}_T + 2\ln(2/\alpha))} \right) + G\sqrt{T} \right),$$

*where $C := \max\{B, G\}$ and $\widehat{\gamma}_T := \max\{\gamma_T, \widetilde{\gamma}_T\}$.*

**Remark 2.** *The (reward) regret here is the stronger version, i.e., $\mathcal{R}_+(T)$. Compared to the unconstrained case, the regret bound has an additional term $\rho G\sqrt{T}$, which roughly captures the impact of the constraint. As in the unconstrained case, one can plug in different $\gamma_T$ and $\widetilde{\gamma}_T$ to see that both regret and constraint violation are sublinear for commonly used kernels. For example, for an SE kernel, both $\gamma_T$ and $\widetilde{\gamma}_T$ are on the order of $(\ln T)^{d+1}$. Finally, the standard "doubling trick" can be used to design an anytime algorithm (i.e., without the knowledge of $T$) with regret and constraint violation bounds of the same order.*

**Proof Sketch of Theorem 1.** The key step is to establish a sublinear bound on the term $\mathcal{R}_+(T) + \phi \sum_{t=1}^{T} g(x_t)$ for any $\phi \in [0, \rho]$. Then, with convex optimization tools, one can establish regret and constraint violation bound. To this end, By the dual update of CKB, we can first show that $\mathcal{R}_+(T) + \phi \sum_{t=1}^{T} g(x_t) \leq \mathcal{T}_1 + \mathcal{T}_2 + \frac{V}{2}\phi^2 + \frac{1}{2V}TG^2$, where

$$\mathcal{T}_1 := \sum_{t=1}^{T} (\mathbb{E}_{\pi^*}[f(x)] - \phi_t \mathbb{E}_{\pi^*}[g(x)]) - \sum_{t=1}^{T} (\bar{f}_t(x_t) - \phi_t \bar{g}_t(x_t)), \tag{5}$$

$$\mathcal{T}_2 := \sum_{t=1}^{T} (\bar{f}_t(x_t) - f(x_t)) + \phi \sum_{t=1}^{T} (g(x_t) - \bar{g}_t(x_t)). \tag{6}$$

Thus, the analysis for each choice of exploration strategy only differ in how to bound $\mathcal{T}_1 + \mathcal{T}_2$. Under GP-UCB exploration, the term $\mathcal{T}_1 \leq 0$ by the optimism while $\mathcal{T}_2$ can be upper bounded by $\mathcal{T}_2 \leq 2\beta_T \sum_{t=1}^{T} \sigma_{t-1}(x_t) + 2\phi\widetilde{\beta}_T \sum_{t=1}^{T} \widetilde{\sigma}_{t-1}(x_t) = O(\beta_T \sqrt{\gamma_T T} + \phi\widetilde{\beta}_T \sqrt{\widetilde{\gamma}_T T})$, i.e., standard predication error bound. Building on the bound on $\mathcal{T}_1 + \mathcal{T}_2$, one can then show the required results. $\quad\square$

### 3.2 A Sufficient Condition for Provably Efficient Explorations

The above analysis reveals that the key step in obtaining sublinear performance guarantees of Algorithm 1 is to find a sublinear bound on $\mathcal{T}_1 + \mathcal{T}_2$, which depends on the choice of exploration strategies. To go beyond GP-UCB exploration, we will establish a sufficient condition on general exploration strategies (i.e., $\mathcal{A}_f$ and $\mathcal{A}_g$), which guarantees a sublinear bound on $\mathcal{T}_1 + \mathcal{T}_2$ and hence sublinear regret and sublinear constraint violation.

We first present the intuition behind the key components of our sufficient condition. Inspired by [39], it suffices to focus on the following three nice events so as to bound $\mathcal{T}_1 + \mathcal{T}_2$ in (5)-(6):

$E^{est} := \{E_f^{est}(x,t) \cap E_g^{est}(x,t); \forall (x,t)\}, E_t^{conc} := \{E_{f,t}^{conc}(x) \cap E_{g,t}^{conc}(x); \forall x\}, E_t^{anti} := E_{f,t}^{anti} \cap E_{g,t}^{anti}$, in which

$E_f^{est}(x,t) := |f(x) - \mu_{t-1}(x)| \leq c_{f,t}^{(1)}\sigma_{t-1}(x), E_g^{est}(x,t) := |g(x) - \widetilde{\mu}_{t-1}(x)| \leq c_{f,t}^{(1)}\widetilde{\sigma}_{t-1}(x)$,

$E_{f,t}^{conc}(x) := |f_t(x) - \mu_{t-1}(x)| \leq c_{f,t}^{(2)}\sigma_{t-1}(x), E_{g,t}^{conc}(x) := |g_t(x) - \widetilde{\mu}_{t-1}(x)| \leq c_{g,t}^{(2)}\widetilde{\sigma}_{t-1}(x)$,

$E_{f,t}^{anti} := \mathbb{E}_{\pi^*}[f_t(x) - \mu_{t-1}(x)] \geq c_{f,t}^{(1)}\mathbb{E}_{\pi^*}[\sigma_{t-1}(x)], E_{g,t}^{anti} := \mathbb{E}_{\pi^*}[g_t(x) - \widetilde{\mu}_{t-1}(x)] \leq -c_{g,t}^{(1)}\mathbb{E}_{\pi^*}[\widetilde{\sigma}_{t-1}(x)]$.

To see the intuition, first suppose that events $E^{est}$ and $E_t^{conc}$ hold with high probability. Then, it is easy to see that the estimates are close to the true functions, and hence, one can derive a bound on $\mathcal{T}_2$ in (6). Now, suppose that events $E^{est}$ and $E_t^{anti}$ hold with some positive probability. Then, one can see that the estimates are optimistic with respect to the true functions when evaluated at the optimal points. This probabilistic optimism is the key to bounding $\mathcal{T}_1$ in (5). Note that GP-UCB exploration is optimistic with probability one by definition (see Definition 1), and hence, $\mathcal{T}_1 \leq 0$ always holds.

To formally state our result, let us define the filtration $\mathcal{F}_t$ as all the history up to the end of round $t$. Let $\mathbb{E}_t[\cdot] := \mathbb{E}[\cdot|\mathcal{F}_{t-1}]$ and $\mathbb{P}_t(\cdot) := \mathbb{P}[\cdot|\mathcal{F}_{t-1}]$. Then, we have the following sufficient condition.

**Assumption 3** (Sufficient Condition). *The sufficient condition includes two parts:*

*(1) (Probability condition)* $\mathbb{P}(E^{est}) \geq 1 - p_1$, $\mathbb{P}_t(E_t^{conc}) \geq 1 - p_{2,t}$, *and* $\mathbb{P}_t(E_t^{anti}) \geq p_3 > 0$ *for some time-dependent sequences* $c_{f,t}^{(1)}, c_{g,t}^{(1)}, c_{f,t}^{(2)},$ *and* $c_{g,t}^{(2)}$.

*(2) (Boundedness condition) (i) There exists a positive probability* $p_4$ *such that* $1 + \frac{2}{(p_3 - p_{2,t})} \leq 1/p_4$ *for all* $t$; *(ii) There exist some functions of* $T$: $c_f(T), c_g(T)$ *such that* $c_{f,t}^{(1)} + c_{f,t}^{(2)} \leq c_f(T)$ *and* $c_{g,t}^{(1)} + c_{g,t}^{(2)} \leq c_g(T)$ *for all* $t$; *(iii)* $\sum_{t=1}^{T} p_{2,t} \leq C'$ *for some constant* $C'$.

**Remark 3.** *The above sufficient condition generalizes existing similar results [39–41] in several aspects. First, existing works mainly focus on the MAB or linear bandit settings, which are special cases of our KB setting (e.g., choosing a linear kernel leads to linear bandit). Second, while existing works only establish bounds on the expected regret, we aim to establish a high-probability bound. As a result, we need the additional boundedness condition, which, however, is simply for technical reasons. Third, in contrast to existing works that consider the unconstrained case only, we consider the constrained case, which is more challenging. Specifically, it requires that $E_t^{anti}$ holds under policy $\pi^*$ (i.e., the expectation over $\pi^*$) rather than under a single optimal action $x^*$.*

With the above sufficient condition, we have the following general performance bounds for CKB.

**Theorem 2.** *Suppose $\rho \geq 4B/\delta$ and $V = G\sqrt{T}/\rho$. Let $\kappa := B + \rho G$. Assume that CKB is equipped with exploration strategies that satisfy the sufficient condition in Assumption 3. Then, under Slater's condition in Assumption 1 and regularity assumptions in Assumption 2, CKB achieves the following bounds on regret and constraint violation with probability at least $1 - \alpha - p_1$ for any $\alpha \in (0, 1)$:*

$$\mathcal{R}_+(T) = O\left(\frac{1}{p_4}c_f(T)\sqrt{T\gamma_T} + \frac{1}{p_4}\rho c_g(T)\sqrt{T\widetilde{\gamma}_T} + \rho G\sqrt{T} + \kappa\frac{c_f(T) + \rho c_g(T)}{p_4}\sqrt{2T\ln(1/\alpha)}\right),$$

$$\mathcal{V}(T) = O\left(\frac{1}{\rho p_4}c_f(T)\sqrt{T\gamma_T} + \frac{1}{p_4}c_g(T)\sqrt{T\widetilde{\gamma}_T} + G\sqrt{T} + \kappa\frac{c_f(T) + \rho c_g(T)}{\rho p_4}\sqrt{2T\ln(1/\alpha)}\right).$$

In the following, we will show that Theorem 2 provides a unified view of the regret and constraint violation performance for various CKB algorithms, thanks to our sufficient condition. In particular, we first show that existing exploration strategies, such as GP-UCB and GP-TS, satisfy our sufficient condition, given by the following corollary.

**Corollary 1.** *Let $\beta_t = B + R\sqrt{2(\gamma_{t-1} + 1 + \ln(1/\alpha_f))}$ and $\widetilde{\beta}_t = B + R\sqrt{2(\widetilde{\gamma}_{t-1} + 1 + \ln(1/\alpha_g))}$.*
*(i). GP-UCB with $\widehat{\beta}_t = \beta_t$ and $\widehat{\beta}_t = -\widetilde{\beta}_t$ for $\mathcal{A}_f$ and $\mathcal{A}_g$, respectively, satisfies the sufficient condition.*

*(ii). GP-TS with $\widehat{\beta}_t = \beta_t$ and $\widehat{\beta}_t = -\widetilde{\beta}_t$ for $\mathcal{A}_f$ and $\mathcal{A}_g$, respectively, satisfies the sufficient condition when $\pi^*$ is a deterministic policy.*

The sufficient condition also enables us to design CKB algorithms with new exploration strategies. In the following, inspired by [40], we propose a new GP-based exploration strategy, which aims to strike a balance between GP-UCB and GP-TS explorations.

**Definition 2** (RandGP-UCB Exploration). *Suppose that the posterior distribution for a black-box function $h$ in round $t$ is given by $\mathcal{GP}(\widehat{\mu}_{t-1}(\cdot), \widehat{k}_{t-1}(\cdot, \cdot))$. Then, the estimate of $h$ in round $t$ under RandGP-UCB exploration strategy is $h_t(\cdot) = \widehat{\mu}_{t-1}(\cdot) + \widehat{Z}_t\widehat{\sigma}_{t-1}(\cdot)$, where $\widehat{Z}_t \sim \widehat{\mathcal{D}}$ for some distribution $\widehat{\mathcal{D}}$ and $\widehat{\sigma}_{t-1}^2(x) = \widehat{k}_{t-1}(x, x)$ for all $x \in \mathcal{X}$.*

In contrast to GP-UCB, RandGP-UCB replaces the deterministic confidence bound by a randomized one. Compared to GP-TS, RandGP-UCB uses "coupled" noise in the sense that all the actions share the same noise $\widehat{Z}_t$ rather than "decoupled" and correlated noise in GP-TS. This subtle difference will not only help to eliminate the additional factor $\sqrt{\ln(|\mathcal{X}|)}$ in GP-TS due to the use of union bound, but also allow us to deal with a general stochastic $\pi^*$.

**Corollary 2.** *Let $\beta_t, \widetilde{\beta}_t$ be the same as in Corollary 1. RandGP-UCB with $\widehat{\mathcal{D}} = \mathcal{N}(0, \beta_t^2)$ and $\widehat{\mathcal{D}} = \mathcal{N}(0, \widetilde{\beta}_t^2)$ for $\mathcal{A}_f$ and $\mathcal{A}_g$, respectively, satisfies the sufficient condition.*

Thus, one can instantiate CKB with RandGP-UCB exploration to obtain a new algorithm called `CKB-Rand` with performance guarantees given by Theorem 2. Note that RandGP-UCB with other distributions $\widehat{\mathcal{D}}$ can also satisfy the sufficient condition (as discussed in Appendix C).

## 4 Discussions

In this section, we discuss several possible questions one may have at this moment about our CKB algorithm. The first natural question to ask is whether one can further improve the constraint violation bound. It turns out that with a minor modification of CKB algorithm, one can achieve a bounded and even zero constraint violation by trading the regret slightly (but still the same order as before). The modification is to introduce a slackness given by $\varepsilon$ in the dual update in Algorithm 1, i.e., $\phi_{t+1} = \text{Proj}_{[0,\rho]}\left[\phi_t + \frac{1}{V}(\bar{g}_t(x_t) + \varepsilon)\right]$ with $\varepsilon \leq \delta/2$. Intuitively speaking, this can be viewed as if one is working on a new pessimistic constraint function. After obtaining the constraint violation under this new hypothetic constraint, one can subtract $\varepsilon T$ to find the true constraint violation under the true function $g$. The catch here is that one needs also change the baseline problem to the following one: $\max_\pi \{\mathbb{E}_\pi[f(x)] : \mathbb{E}_\pi[g(x)] + \varepsilon \leq 0\}$ so that it matches the new pessimistic constraint function. Let $\pi_\varepsilon^*$ be the optimal solution to this new problem and the obtained regret is only with respect to $\pi_\varepsilon^*$ rather than the original $\pi^*$. Thus, we need to further bound the following difference $T\mathbb{E}_{\pi^*}[f(x)] - T\mathbb{E}_{\pi_\varepsilon^*}[f(x)]$ to obtain the true regret bound, which is given in Appendix E.

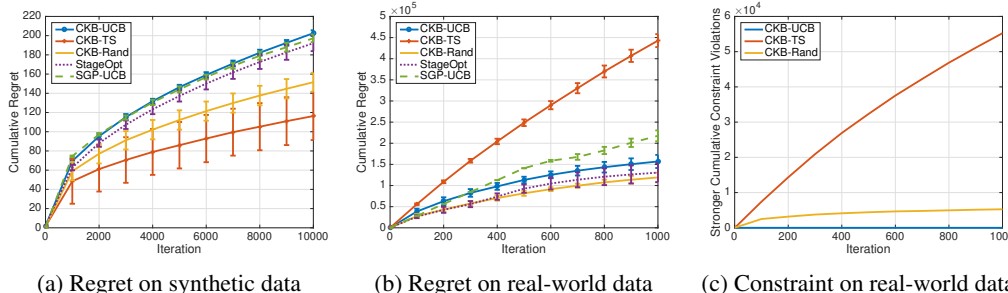

| (a) Regret on synthetic data | (b) Regret on real-world data | (c) Constraint on real-world data |

Figure 1: Experimental results on constrained kernelized bandits with light-tailed data .

Another question one may ask is why we are able to go beyond UCB-type exploration compared to existing works (e.g., [13–15]) that only establish performance guarantees under UCB. The answer is that they resort to another popular method for constrained optimization based on the so-called Lyapunov-drift argument. However, this method relies heavily on the optimism under UCB to derive both regret and constraint violation bounds. Thus, it is difficult to generalize it for other exploration strategies even for simpler linear bandit problems. In contrast, our analysis relies only on the bound for $\mathcal{T}_1 + \mathcal{T}_2$, which allows us to consider more general exploration strategies. In Appendix F, we also discuss how to apply the Lyapunov-drift based method to analyze UCB exploration in our setting, through which we highlight more differences and some subtleties in Lyapunov-drift method.

## 5 Simulation Results

Although our work is mainly a theoretical study, we conduct simulations to compare the performance of our algorithms (i.e., `CKB-UCB`, `CKB-TS`, and `CKB-Rand`, that is, CKB with GP-UCB, GP-TS, and RandGP-UCB explorations, respectively) with existing safe KB algorithms based on both synthetic and real-world datasets. In particular, we consider the two most recent safe KB algorithms: `StageOpt` [28] (which has a superior performance compared to `SafeOpt` [27]) and `SGP-UCB` [3]. Our goal is to show that our proposed CKB algorithms can trade a slight performance in constraint violation (i.e., soft constraint) for improvement in the reward regret, computation complexity and flexible implementations, i.e., a better complexity-regret-constraint trade-off.

### 5.1 Synthetic Data and Light-Tailed Real-World Data

**Synthetic Data.** The domain $\mathcal{X}$ is generated by discretizing $[0, 1]$ uniformly into 100 points. The objective function $f(\cdot) = \sum_{i=1}^{p} a_i k(\cdot, x_i)$ is generated by uniformly sampling $a_i \in [-1, 1]$ and support points $x_i \in \mathcal{X}$ with $p = 100$. With the same manner, we generate the constraint function $g$. The kernel is $k_{se}$ with parameter $l = 0.2$. Other parameters include $B$, $R$ and $\gamma_t$ are set similar as in the unconstrained case (e.g., [5]).

**Light-Tailed Real-World Data.** We use the light sensor data collected in the CMU Intelligent Workplace in Nov 2005, which is available online as Matlab structure[3] and contains locations of 41 sensors, 601 train samples and 192 test samples. We use it in the context of finding the maximum average reading of the sensors. In particular, $f$ is set as empirical average of the test samples, with $B$ set as its maximum, and $k$ is set as the empirical covariance of the normalized train samples. The constraint is given by $g(\cdot) = -f(\cdot) + h$ with $h = B/2$. We perform 50 trials (each with $T = 10,000$) and plot the mean of the cumulative regret along with the error bars, as shown in Fig. 1.

**Regret.** Our three CKB algorithms achieve a better (or similar) regret performance compared to the existing safe BO algorithms (see Figures 1(a) and 1(b)). Among the three CKB algorithms, `CKB-Rand` appears to have reasonably good performance at all times.

**Constraint violation.** Since we have $\mathcal{V}(T) = 0$ under all the algorithms, we study the total number of rounds where the constraint is violated, denoted by $N$. In the synthetic data setting, our proposed CKB algorithms have $N \leq 5$ over $T = 10,000$ rounds; in the real-world data setting, `CKB-UCB` enjoys $N = 0$ and `CKB-Rand` has an average $N = 38$ over a horizon $T = 1,000$. Furthermore, we plot the stronger cumulative constraint violations given by $\sum_{t=1}^{T} [g(x_t)]_+$ as shown in Figure 1(c),

---

[3] http://www.cs.cmu.edu/~guestrin/Class/10708-F08/projects/

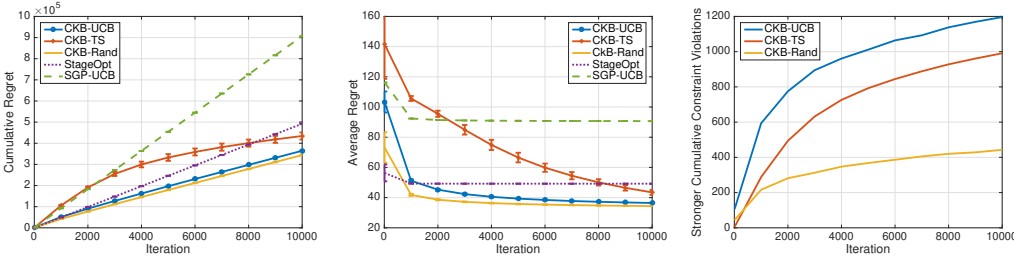

(a) Regret on finance data     (b) Avg. regret on finance data     (c) Constraint on finance data

Figure 2: Experimental results on constrained kernelized bandits under heavy-tailed finance data.

from which we can see that all CKB algorithms achieve sublinear performance even with respect to this stronger metric, i.e., violation cancellation across rounds is not allowed.

**Practical considerations.** Our proposed CKB algorithms have the same computational complexity as the unconstrained case. In particular, they scale linearly with the number of actions in the discrete-domain case[4]. On the other hand, `StageOpt` scales quadratically due to the construction of the safe set, and `SGP-UCB` requires the additional random initialization stage, which leads to linear regret at the beginning of the learning process. Moreover, standard methods for improving the scalability of unconstrained KB can be naturally applied to our CKB algorithms [42]. Finally, both `StageOpt` and `SGP-UCB` require the knowledge of a safe action (i.e., one that satisfies the constraint) in advance, and moreover, `StageOpt` requires $f$ to be Lipschitz and needs to estimate the Lipschitz constant, which impacts the robustness. In contrast, CKB algorithms for soft constraints only require a mild Slater's condition as in Assumption 1, which does not require the existence of a safe action.

### 5.2 Heavy-Tailed Real-World Data

We further compare different constrained KB algorithms in a new real-world dataset, which demonstrates a heavy-tailed noise. Note that sub-Gaussian noise is required in all the existing theoretical works (including our work). We use this dataset to test the robustness of various constrained KB algorithms. The experimental results tend to show that our three CKB algorithms are more robust in terms of heavy-tailed noise, which is common in practical applications. The detail of this real-world dataset is deferred to Appendix D.

**Regret.** We plot both cumulative regret and time-average regret in this setting (see Figures 2 (a) and (b)). We can observe that in the presence of heavy-tailed noise, our three CKB algorithms have significant performance gain over existing safe KB algorithms.

**Constraint violation.** We focus on the strong metric, i.e., the number of rounds where the constraint is violated, denoted by $N$. We have that `CKB-Rand` enjoys an average $N = 21$ and `CKB-UCB` has an average $N = 47$ within the horizon of $T = 10,000$. We also plot stronger cumulative constraint violations given by $\sum_{t=1}^{T}[g(x_t)]_+$ as shown in Figure 2(c), from which we can see that all CKB algorithms achieve sublinear performance even with respect to this stronger metric.

## 6 Conclusion

We presented a general framework for constrained KB with soft constraints via primal-dual optimization. Armed with our developed sufficient condition, this framework not only allows us to design provably efficient (i.e., sublinear reward regret and sublinear total constraint violation) CKB algorithms with both UCB and TS explorations, but presents a unified method to design new effective ones. By introducing slackness, our algorithm can also attain a bounded or even zero constraint violation while still achieving a sublinear regret. We further perform simulations on both synthetic data and real-world data that corroborate our theoretical results. Along the way, we also present the first detailed discussion on two existing methods for analyzing constrained bandits and MDPs by highlighting interesting insights. For future work, one interesting direction is to build upon recent advances in unconstrained KB that attain an improved regret (e.g., [43–46]) to study the corresponding constrained case in the hope to maintain the same improvement.

---

[4]For a continuous domain, as in the unconstrained case, one can resort to heuristic solvers (e.g., a combination of random sampling (cheap) and the "L-BFGS-B'" optimization method.). In fact, to attain the same order of regret bound, the solution to the acquisition maximization problem need not be exact. Instead, it only needs to maximize the acquisition function within $C/\sqrt{t}$ accuracy for some constant $C$ at each step.

## Acknowledgement

We would like to thank the anonymous reviewers for their helpful comments. XZ is partially supported by NSF grant CNS-2153220. BJ is partially supported by NSF grant CNS-2112694.

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
