# Appendix

## A   Proof of Theorem 1

Before we present the proof, we first obtain the following lemma on the dual variable.

**Lemma 1.** *Under the update rule of $\phi_t$ in Algorithm 1, we have for any $\phi \in [0, \rho]$,*

$$\sum_{t=1}^{T} \bar{g}_t(x_t)(\phi - \phi_t) \leq \frac{V}{2}(\phi_1 - \phi)^2 + \sum_{t=1}^{T} \frac{1}{2V} \bar{g}_t(x_t)^2.$$

*Proof.* By the dual variable update rule in Algorithm 1 and the non-expansiveness of projection to $[0, \rho]$, we have

$$(\phi_{t+1} - \phi)^2 \leq (\phi_t + \frac{1}{V} \bar{g}_t(x_t) - \phi)^2$$

$$= (\phi_t - \phi)^2 + \frac{2}{V} \bar{g}_t(x_t)(\phi_t - \phi) + \frac{1}{V^2} \bar{g}_t(x_t)^2.$$

Summing over $T$ steps and multiplying both sides by $\frac{V}{2}$, we have

$$\frac{V}{2}(\phi_{T+1} - \phi)^2 - \frac{V}{2}(\phi_1 - \phi)^2 \leq \sum_{t=1}^{T} \bar{g}_t(x_t)(\phi_t - \phi) + \sum_{t=1}^{T} \frac{1}{2V} \bar{g}_t(x_t)^2.$$

Hence,

$$\sum_{t=1}^{T} \bar{g}_t(x_t)(\phi - \phi_t) \leq \frac{V}{2}(\phi_1 - \phi)^2 + \sum_{t=1}^{T} \frac{1}{2V} \bar{g}_t(x_t)^2, \tag{7}$$

which completes the proof. $\square$

Now, we are ready to present the proof of Theorem 1.

*Proof of Theorem 1.* Under Slater condition in Assumption 1, we have the boundedness of the optimal dual solution by standard convex optimization analysis (cf. [38, Theorem 8.42])

$$0 \leq \phi^* \leq \frac{(\mathbb{E}_{\pi^*}[f(x)] - \mathbb{E}_{\pi_0}[f(x)])}{\delta} \leq \frac{2B}{\delta},$$

where the last inequality holds by the boundedness of $f(x)$. Note that the reason why we can use convex analysis is that $\mathbb{E}_\pi[h(x)]$ for any fixed $h$ is a linear function with respect to $\pi$ (and is thus convex). Now, we turn to establish a bound over $\mathcal{R}_+(T) + \phi \sum_{t=1}^{T} g(x_t)$. First, note that

$$\mathcal{R}_+(T) + \phi \sum_{t=1}^{T} g(x_t)$$

$$= T\mathbb{E}_{\pi^*}[f(x)] - \sum_{t=1}^{T} f(x_t) + \phi \sum_{t=1}^{T} g(x_t)$$

$$= T\mathbb{E}_{\pi^*}[f(x)] - \sum_{t=1}^{T} \bar{f}_t(x_t) + \sum_{t=1}^{T} \bar{f}_t(x_t) - f(x_t) + \phi \sum_{t=1}^{\tau} g(x_t). \tag{8}$$

We can further bound (8) by using Lemma 1. In particular, we have

$$T\mathbb{E}_{\pi^*}[f(x)] - \sum_{t=1}^{T}\bar{f}_t(x_t) + \sum_{t=1}^{T}\bar{f}_t(x_t) - f(x_t) + \phi\sum_{t=1}^{\tau}g(x_t)$$

$$\overset{(a)}{\leq} \sum_{t=1}^{T}\mathbb{E}_{\pi^*}[f(x)] - \phi_t\mathbb{E}_{\pi^*}[g(x)] - \sum_{t=1}^{T}\bar{f}_t(x_t) + \sum_{t=1}^{T}\bar{f}_t(x_t) - f(x_t) + \phi\sum_{t=1}^{T}g(x_t)$$

$$\overset{(b)}{=} \sum_{t=1}^{T}\mathbb{E}_{\pi^*}[f(x)] - \phi_t\mathbb{E}_{\pi^*}[g(x)] - \left(\sum_{t=1}^{T}\bar{f}_t(x_t) - \phi_t\bar{g}_t(x_t)\right) + \sum_{t=1}^{T}\bar{f}_t(x_t) - f(x_t)$$

$$+ \phi\left(\sum_{t=1}^{T}g(x_t) - \bar{g}_t(x_t)\right) + \sum_{t=1}^{T}\bar{g}_t(x_t)(\phi - \phi_t)$$

$$\overset{(c)}{\leq} \sum_{t=1}^{T}\mathbb{E}_{\pi^*}[f(x)] - \phi_t\mathbb{E}_{\pi^*}[g(x)] - \left(\sum_{t=1}^{T}\bar{f}_t(x_t) - \phi_t\bar{g}_t(x_t)\right) + \sum_{t=1}^{\tau}\bar{f}_t(x_t) - f(x_t)$$

$$+ \phi\left(\sum_{t=1}^{T}g(x_t) - \bar{g}_t(x_t)\right) + \frac{V}{2}(\phi_1 - \phi)^2 + \sum_{t=1}^{T}\frac{1}{2V}\bar{g}_t(x_t)^2$$

$$\overset{(d)}{=} \mathcal{T}_1 + \mathcal{T}_2 + \frac{V}{2}\phi^2 + \frac{1}{2V}TG^2, \tag{9}$$

where (a) holds since $\phi_t \geq 0$ and $\mathbb{E}_{\pi^*}[g(x)] \leq 0$; (b) holds by adding and subtracting terms; (c) follow from Lemma 1 to bound the last term; (d) holds by the fact $\phi_1 = 0$, the boundedness of $\bar{g}_t$ and the definitions of $\mathcal{T}_1$ and $\mathcal{T}_2$, i.e.,

$$\mathcal{T}_1 = \sum_{t=1}^{T}(\mathbb{E}_{\pi^*}[f(x)] - \phi_t\mathbb{E}_{\pi^*}[g(x)]) - \sum_{t=1}^{T}(\bar{f}_t(x_t) - \phi_t\bar{g}_t(x_t)), \tag{10}$$

$$\mathcal{T}_2 = \sum_{t=1}^{T}(\bar{f}_t(x_t) - f(x_t)) + \phi\sum_{t=1}^{T}(g(x_t) - \bar{g}_t(x_t)). \tag{11}$$

Plugging (9) into (8), yields for any $\phi \in [0, \rho]$,

$$\mathcal{R}_+(T) + \phi\sum_{t=1}^{T}g(x_t) \leq \mathcal{T}_1 + \mathcal{T}_2 + \frac{V}{2}\phi^2 + \frac{1}{2V}TG^2. \tag{12}$$

First, assume that we already have a bound on $\mathcal{T}_1 + \mathcal{T}_2$, i.e., $\mathcal{T}_1 + \mathcal{T}_2 \leq \chi(T, \phi)$ with high probability, and $\chi(T, \phi)$ is an increasing function in $\phi$. This directly leads to the following inequality (with $V = G\sqrt{T}/\rho$) for any $\phi \in [0, \rho]$:

$$\mathcal{R}_+(T) + \phi\sum_{t=1}^{T}g(x_t) \leq \chi(T, \phi) + \frac{\phi^2 G\sqrt{T}}{2\rho} + \frac{\rho G\sqrt{T}}{2}. \tag{13}$$

Based on this key inequality, we can analyze both regret and constraint violation.

**Regret**. We can simply choose $\phi = 0$ in (13), and obtain that with high probability

$$\mathcal{R}_+(T) = O\left(\chi(T, 0) + \rho G\sqrt{T}\right). \tag{14}$$

**Constraint violation**. To obtain the bound on $\mathcal{V}(T)$, inspired by [11], we will resort to tools from constrained convex optimization. First, we have $\frac{1}{T}\sum_{t=1}^{T}f(x_t) = \mathbb{E}_{\pi'}[f(x)]$ and $\frac{1}{T}\sum_{t=1}^{T}g(x_t) = \mathbb{E}_{\pi'}[g(x)]$ for some probability measure $\pi'$ by the convexity of probability measure. As a result, we have

$$\mathbb{E}_{\pi^*}[f(x)] - \mathbb{E}_{\pi'}[f(x)] + \rho\left[\mathbb{E}_{\pi'}[g(x)]\right]_+ = \frac{1}{T}\mathcal{R}_+(T) + \frac{1}{T}\phi\sum_{t=1}^{T}g(x_t) \leq \frac{\chi(T, \rho) + \rho G\sqrt{T}}{T}, \tag{15}$$

where $[a]_+ := \max\{0, a\}$, and the first equality holds by choosing $\phi = \rho$ if $\sum_{t=1}^T g(x_t) \geq 0$, and otherwise $\phi = 0$, and the second inequality holds by upper bounding RHS of (13) with $\phi = \rho$ since (13) holds for all $\phi \in [0, \rho]$ and $\chi(T, \phi)$ is increasing in $\phi$.

Then, we will apply the following useful lemma, which is adapted from Theorem 3.60 in [38].

**Lemma 2.** *Consider the following convex constrained problem $h(\pi^*) = \max_{\pi \in \mathcal{C}}\{h(\pi) : w(\pi) \leq 0\}$, where both $h$ and $w$ are convex over the convex set $\mathcal{C}$ in a vector space. Suppose $h(\pi^*)$ is finite and there exists a slater point $\pi_0$ such that $w(\pi_0) \leq -\delta$, and a constant $\rho \geq 2\kappa^*$, where $\kappa^*$ is the optimal dual variable, i.e., $\kappa^* = \operatorname{argmin}_{\lambda \geq 0}(\max_\pi h(\pi) - \kappa w(\pi))$. Assume that $\pi' \in \mathcal{C}$ satisfies*

$$h(\pi^*) - h(\pi') + \rho [w(\pi')]_+ \leq \varepsilon, \tag{16}$$

*for some $\varepsilon > 0$, then we have $[w(\pi')]_+ \leq 2\varepsilon/\rho$.*

Thus, since (15) satisfies (16) and $\mathbb{E}_\pi [h(x)]$ for any fixed $h$ is a linear function with respect to $\pi$, by Lemma 2, we have

$$\mathcal{V}(T) = O\left(\frac{1}{\rho}\chi(T, \rho) + G\sqrt{T}\right). \tag{17}$$

We are only left to bound $\mathcal{T}_1 + \mathcal{T}_2$ by $\chi(T, \phi)$. To this end, we will resort to standard concentration results for GP bandits. First, by [5, Theorem 2], we have the following lemma.

**Lemma 3.** *Fix $\alpha \in (0, 1]$, with probability at least $1 - \alpha$, the followings hold simultaneously for all $t \in [T]$ and all $x \in \mathcal{X}$*

$$|f(x) - \mu_{t-1}(x)| \leq \beta_t \sigma_{t-1}(x), \quad |g(x) - \widetilde{\mu}_{t-1}(x)| \leq \widetilde{\beta}_t \widetilde{\sigma}_{t-1}(x),$$

Thus, based on this lemma and the definition of GP-UCB exploration, we have with high probability, $f_t(x) \geq f(x)$ and $g_t(x) \leq g(x)$ for all $t \in [T]$ and $x \in \mathcal{X}$. This directly implies that $\bar{f}_t(x) \geq f(x)$ and $\bar{g}_t(x) \leq g(x)$ for all $t \in [T]$ and $x \in \mathcal{X}$ (i.e., optimistic estimates), which holds by $|f(x)| \leq B$ and $|g(x)| \leq G$ and the way of truncation in Algorithm 1. Now, to bound $\mathcal{T}_1$ in (10), we have

$$\mathcal{T}_1 = \sum_{t=1}^T (\mathbb{E}_{\pi^*}[f(x)] - \mathbb{E}_{\pi^*}[\bar{f}_t(x)] + \mathbb{E}_{\pi^*}[\bar{f}_t(x)] - \bar{f}_t(x_t))$$

$$+ \phi_t \sum_{t=1}^T (\bar{g}_t(x_t) - \mathbb{E}_{\pi^*}[\bar{g}_t(x)] + \mathbb{E}_{\pi^*}[\bar{g}_t(x)] - \mathbb{E}_{\pi^*}[g(x)])$$

$$\stackrel{(a)}{\leq} \sum_{t=1}^T (\mathbb{E}_{\pi^*}[\bar{f}_t(x)] - \bar{f}_t(x_t)) + \phi_t \sum_{t=1}^T (\bar{g}_t(x_t) - \mathbb{E}_{\pi^*}[\bar{g}_t(x)])$$

$$= \sum_{t=1}^T (\mathbb{E}_{\pi^*}[\bar{f}_t(x)] - \phi_t \mathbb{E}_{\pi^*}[\bar{g}_t(x)] - (\bar{f}_t(x_t) - \phi_t \bar{g}_t(x_t)))$$

$$\stackrel{(b)}{\leq} 0,$$

where (a) holds by the fact that estimates are optimistic, i.e., $\bar{f}_t(x) \geq f(x)$ and $\bar{g}_t(x) \leq g(x)$ for all $t \in [T]$ and $x \in \mathcal{X}$; (b) holds by the greedy selection of Algorithm 1.

Now, we turn to bound $\mathcal{T}_2$. In particular, we have

$$\mathcal{T}_2 \stackrel{(a)}{\leq} \sum_{t=1}^T 2\beta_t \sigma_{t-1}(x_t) + \phi \sum_{t=1}^T 2\widetilde{\beta}_t \widetilde{\sigma}_{t-1}(x_t)$$

$$\stackrel{(b)}{\leq} O\left(\beta_T \sqrt{T\gamma_T} + \phi \widetilde{\beta}_T \sqrt{T\widetilde{\gamma}_T}\right), \tag{18}$$

where (a) holds by Lemma 3 and the definition of GP-UCB exploration, i.e., $f_t(x) = \mu_{t-1}(x) + \beta_t \sigma_{t-1}(x)$ and $g_t(x) = \widetilde{\mu}_{t-1}(x) - \widetilde{\beta}_t \widetilde{\sigma}_{t-1}(x)$. Note that truncation also does not affect this step; (b) holds by Cauchy-Schwartz inequality and the bound of sum of predictive variance (cf. [5, Lemma 4]). Note that we have also used the fact that $\beta_t$ and $\widetilde{\beta}_t$ is increasing in $t$.

Putting the bounds on $\mathcal{T}_1$ and $\mathcal{T}_2$ together, we have obtained that with high probability

$$\mathcal{T}_1 + \mathcal{T}_2 \le \chi(T, \phi) := O\left(\beta_T \sqrt{T\gamma_T} + \phi\widetilde{\beta}_T\sqrt{T\widetilde{\gamma}_T}\right).$$

Finally, plugging $\chi(T, 0)$ into (14), yield the regret bound as follows (note that $\beta_t = B + R\sqrt{2(\gamma_{t-1} + 1 + \ln(2/\alpha))}$)

$$\mathcal{R}_+(T) = O\left(B\sqrt{T\gamma_T} + \sqrt{T\gamma_T(\gamma_T + \ln(2/\alpha))} + \rho G\sqrt{T}\right),$$

and plugging $\chi(T, \rho)$ into (17), yields the bound on constraint violation as

$$\mathcal{V}(T) = O\left(\left(1 + \frac{1}{\rho}\right)\left(C\sqrt{T\widehat{\gamma}_T} + \sqrt{T\widehat{\gamma}_T(\widehat{\gamma}_T + \ln(2/\alpha))}\right) + G\sqrt{T}\right),$$

where $C := \max\{B, G\}$ and $\widehat{\gamma}_T := \max\{\gamma_T, \widetilde{\gamma}_T\}$. Hence, it completes the proof. $\qquad\square$

## B  Proof of Theorem 2

Before we present the proof, we introduce a new notation to make the presentation easier. In particular, we let $h(\pi) := \mathbb{E}_\pi[h(x)]$ for any function $h$ and $\pi_t$ is a dirac delta function at the point $x_t$.

*Proof of Theorem 2.* As shown in the proof of Theorem 1, all we need to do is to find a high probability bound over $\mathcal{T}_1 + \mathcal{T}_2$ under the sufficient condition in Assumption 3. Under our newly introduced notation, we have

$$\mathcal{T}_1 + \mathcal{T}_2 = \sum_{t=1}^{T}\left(z_{\phi_t}(\pi^*) - \widehat{z}_{\phi_t}(\pi_t) + \widehat{z}_\phi(\pi_t) - z_\phi(\pi_t)\right) := \sum_{t=1}^{T} d_t, \qquad (19)$$

where $z_{\phi_t}(\cdot) := f(\cdot) - \phi_t g(\cdot)$ and $\widehat{z}_{\phi_t}(\cdot) := \bar{f}_t(\cdot) - \phi_t \bar{g}_t(\cdot)$, and similar definitions for $z_\phi$ and $\widehat{z}_\phi$.

Let $\Delta_{\phi_t}(\pi) := z_{\phi_t}(\pi^*) - z_{\phi_t}(\pi) = (f(\pi^*) - \phi_t g(\pi^*)) - (f(\pi) - \phi_t g(\pi))$. Then, we define the 'undersampled' set as

$$\bar{S}_t := \{\pi \in \Pi : \alpha_{\phi_t}(\pi) := c_{f,t}\sigma_{t-1}(\pi) + \phi_t c_{g,t}\widetilde{\sigma}_{t-1}(\pi) \ge \Delta_{\phi_t}(\pi)\},$$

where $c_{f,t} = (c_{f,t}^{(1)} + c_{f,t}^{(2)})$ and $c_{g,t} = (c_{g,t}^{(1)} + c_{g,t}^{(2)})$ (similarly $\alpha_\phi(\pi) := c_{f,t}\sigma_{t-1}(\pi) + \phi c_{g,t}\widetilde{\sigma}_{t-1}(\pi)$).
Let $u_t = \arg\min_{\pi \in \bar{S}_t} \alpha_{\phi_t}(\pi)$. Thus, conditioned on $E^{est}$ and $E_t^{conc}$, we have

$$\begin{aligned}
d_t &= z_{\phi_t}(\pi^*) - \widehat{z}_{\phi_t}(\pi_t) + \widehat{z}_\phi(\pi_t) - z_\phi(\pi_t) \\
&= z_{\phi_t}(\pi^*) - z_{\phi_t}(u_t) + z_{\phi_t}(u_t) - \widehat{z}_{\phi_t}(\pi_t) + \widehat{z}_\phi(\pi_t) - z_\phi(\pi_t) \\
&= \Delta_{\phi_t}(u_t) + z_{\phi_t}(u_t) - \widehat{z}_{\phi_t}(\pi_t) + \widehat{z}_\phi(\pi_t) - z_\phi(\pi_t) \\
&\overset{(a)}{\le} \Delta_{\phi_t}(u_t) + \widehat{z}_{\phi_t}(u_t) - \widehat{z}_{\phi_t}(\pi_t) + \alpha_{\phi_t}(u_t) + \alpha_\phi(\pi_t) \\
&\overset{(b)}{\le} \Delta_{\phi_t}(u_t) + \alpha_{\phi_t}(u_t) + \alpha_\phi(\pi_t) \\
&\overset{(c)}{\le} 2\alpha_{\phi_t}(u_t) + \alpha_\phi(\pi_t), \qquad (20)
\end{aligned}$$

where (a) holds since under event $E^{est} \cap E_t^{conc}$, for all $x$, $|f(x) - f_t(x)| \le (c_{f,t}^{(1)} + c_{f,t}^{(2)})\sigma_{t-1}(x)$ and $|g(x) - g_t(x)| \le (c_{g,t}^{(1)} + c_{g,t}^{(2)})\widetilde{\sigma}_{t-1}(x)$ and the facts that $|g(x) - \bar{g}_t(x)| \le |g(x) - g_t(x)|$ since $|g(x)| \le G$ and $|f(x) - \bar{f}_t(x)| \le |f(x) - f_t(x)|$ since $|f(x)| \le B$; (b) holds by the greedy selection

in Algorithm 1; (c) follows from $u_t \in \bar{S}_t$. Thus, conditioned on $E^{est}$, we have

$$\mathbb{E}_t \left[ d_t \right] = \mathbb{E}_t \left[ d_t I\{E_t^{conc}\} \right] + \mathbb{E}_t \left[ d_t I\{\bar{E}_t^{conc}\} \right]$$

$$\overset{(a)}{\leq} \mathbb{E}_t \left[ r_t I\{E_t^{conc}\} \right] + (4B + 4\rho G)p_{2,t}$$

$$\overset{(b)}{\leq} \mathbb{E}_t \left[ \alpha_\phi(\pi_t) \right] + 2\alpha_{\phi_t}(u_t) + (4B + 4\rho G)p_{2,t}$$

$$\overset{(c)}{\leq} \mathbb{E}_t \left[ \alpha_\phi(\pi_t) \right] + 2\frac{\mathbb{E}_t \left[ \alpha_{\phi_t}(\pi_t) \right]}{\mathbb{P}_t \left( x_t \in \bar{S}_t \right)} + (4B + 4\rho G)p_{2,t}$$

$$\overset{(d)}{=} \left( 1 + \frac{2}{\mathbb{P}_t \left( \pi_t \in \bar{S}_t \right)} \right) \mathbb{E}_t \left[ \alpha_\rho(\pi_t) \right] + (4B + 4\rho G)p_{2,t},$$

where (a) holds by definition of $p_{2,t}$, the fact that $\phi, \phi_t \leq \rho$ and the boundedness of functions; (b) follows from Eq. (20) and the fact that given $\mathcal{F}_{t-1}$, $\alpha_{\phi_t}(u_t)$ is deterministic; (c) holds by the following argument: $\mathbb{E}_t \left[ \alpha_{\phi_t}(\pi_t) \right] \geq \mathbb{E}_t \left[ \alpha_{\phi_t}(\pi_t) | \pi_t \in \bar{S}_t \right] \mathbb{P}_t \left( \pi_t \in \bar{S}_t \right) \geq \alpha_{\phi_t}(u_t)\mathbb{P}_t \left( \pi_t \in \bar{S}_t \right)$, which holds by the definition of $u_t$ and the fact that $\alpha_{\phi_t}(u_t)$ and $S_t$ are both $\mathcal{F}_{t-1}$-measurable; (d) holds by definition $\alpha_\rho(\pi_t) := c_{f,t}\sigma_{t-1}(\pi_t) + \rho c_{g,t}\tilde{\sigma}_{t-1}(\pi_t)$ and the fact that both $\phi, \phi_t$ are bounded by $\rho$. Hence, the key is to find a lower bound on the probability $\mathbb{P}_t \left( \pi_t \in \bar{S}_t \right)$. In particular, conditioned on $E^{est}$, we have

$$\mathbb{P}_t \left( \pi_t \in \bar{S}_t \right)$$

$$\overset{(a)}{\geq} \mathbb{P}_t \left( \hat{z}_{\phi_t}(\pi^*) \geq \max_{\pi_j \in S_t} \hat{z}_{\phi_t}(\pi_j), E_t^{conc} \right)$$

$$\overset{(b)}{\geq} \mathbb{P}_t \left( \hat{z}_{\phi_t}(\pi^*) \geq z_{\phi_t}(\pi^*), E_t^{conc} \right)$$

$$\geq \mathbb{P}_t \left( \hat{z}_{\phi_t}(\pi^*) \geq z_{\phi_t}(\pi^*) \right) - \mathbb{P}_t \left( \bar{E}_t^{conc} \right)$$

$$\geq \mathbb{P}_t \left( \bar{f}_t(\pi^*) \geq f(\pi^*), \bar{g}_t(\pi^*) \leq g(\pi^*) \right) - \mathbb{P}_t \left( \bar{E}_t^{conc} \right)$$

$$\overset{(c)}{=} \mathbb{P}_t \left( f_t(\pi^*) \geq f(\pi^*), g_t(\pi^*) \leq g(\pi^*) \right) - \mathbb{P}_t \left( \bar{E}_t^{conc} \right)$$

$$\overset{(d)}{\geq} \mathbb{P}_t \left( f_t(\pi^*) \geq \mu_{t-1}(\pi^*) + c_{f,t}^{(1)}\sigma_{t-1}(\pi^*), g_t(\pi^*) \leq \tilde{\mu}_{t-1}(\pi^*) - c_{g,t}^{(1)}\tilde{\sigma}_{t-1}(\pi^*) \right) - \mathbb{P}_t \left( \bar{E}_t^{conc} \right)$$

$$= \mathbb{P}_t \left( E_t^{anti} \right) - \mathbb{P}_t \left( \bar{E}_t^{conc} \right)$$

$$= p_3 - p_{2,t},$$

where (a) holds by the greedy selection in Algorithm 1 and $\pi^* \in \bar{S}_t$ since $\Delta_{\phi_t}(\pi^*) = 0$. Note that $S_t$ is the complement of the 'undersampled' set $\bar{S}_t$; (b) holds given $E^{est} \cap E_t^{conc}$, for all $\pi_j \in S_t$ $\hat{z}_{\phi_t}(\pi_j) \leq z_{\phi_t}(\pi_j) + \alpha_{\phi_t}(\pi_j) \leq z_{\phi_t}(\pi_j) + \Delta_{\phi_t}(\pi_j) = z_{\phi_t}(\pi^*)$; (c) holds since $|g(x)| \leq G$ for all $x$ and $|f(x)| \leq B$ for all $x$; (d) holds since under $E^{est}$, we have $f(x) \leq \mu_{t-1}(x^*) + c_{1,f}\sigma_{t-1}(x^*)$ and $g(x) \geq \tilde{\mu}_{t-1}(x^*) - c_{1,g}\tilde{\sigma}_{t-1}(x^*)$ for all $x$.

Putting everything together, we have now arrived at that conditioned on $E^{est}$,

$$\mathbb{E}_t \left[ d_t \right] \leq \mathbb{E}_t \left[ \alpha_\rho(x_t) \right] \left( 1 + \frac{2}{p_3 - p_{2,t}} \right) + (4B + 4\rho G)p_{2,t}$$

$$\leq \frac{1}{p_4}\mathbb{E}_t \left[ \alpha_\rho(x_t) \right] + (4B + 4\rho G)p_{2,t}. \tag{21}$$

where the last inequality follows from the boundedness condition in the sufficient condition. In order to obtain a high probability bound, inspired by [5], we will resort to martingale techniques. Let us define the following terms

**Definition 3.** *Define $Y_0 = 0$, and for all $t = 1, \ldots, T$,*

$$\bar{d}_t = d_t \mathcal{I}\{E^{est}\}$$

$$X_t = \bar{d}_t - \frac{1}{p_4}\alpha_\rho(x_t) - (4B + 4\rho G)p_{2,t}$$

$$Y_t = \sum_{s=1}^{t} X_s,$$

*where $\mathcal{I}\{\cdot\}$ is the indicator function.*

Now, we can show that $\{Y_t\}_t$ is a super-martingale with respect to filtration $\mathcal{F}_t$. To this end, we need to show that for any $t$ and any possible $\mathcal{F}_{t-1}$, $\mathbb{E}\left[Y_t - Y_{t-1}|\mathcal{F}_{t-1}\right] \leq 0$, i.e., $\mathbb{E}_t\left[\bar{d}_t\right] \leq \frac{1}{p_4}\mathbb{E}_t\left[\alpha_\rho(x_t)\right] + (4B + 4\rho G)p_{2,t}$. For $\mathcal{F}_{t-1}$ such that $E^{est}$ holds, we already obtained the required inequality as in Eq. (21). For $\mathcal{F}_{t-1}$ such that $E^{est}$ does not hold, the required inequality trivially holds since the LHS is zero. Now, we turn to show that $\{Y_t\}_t$ is a bounded incremental sequence, i.e., $|Y_t - Y_{t-1}| \leq M_t$ for some constant $M_t$. We first note that

$$|Y_t - Y_{t-1}| = |X_t| \leq |\bar{d}_t| + \frac{1}{p_t}\alpha_\rho(x_t) + (4B + 4\rho G)p_{2,t}$$

$$= |\bar{d}_t| + \frac{1}{p_4}\left(c_{f,t}\sigma_{t-1}(x_t) + \rho c_{g,t}\widetilde{\sigma}_{t-1}(x_t)\right) + (4B + 4\rho G)p_{2,t}$$

$$\overset{(a)}{\leq}(4B + 4\rho G) + \frac{1}{p_4}(c_{f,t} + \rho c_{g,t}) + (4B + 4\rho G)p_{2,t}$$

$$\leq \frac{1}{p_4}(c_{f,t} + \rho c_{g,t})(4B + 4\rho G) := M_t,$$

where (a) holds since $\bar{d}_t \leq d_t \leq (4B + 4\rho G)$, $\sigma_{t-1}(x_t) \leq \sigma_0(x_t) \leq 1$ and $\widetilde{\sigma}_{t-1}(x_t) \leq \widetilde{\sigma}_0(x_t) \leq 1$. Thus, we can apply Azuma-Hoeffding inequality to obtain that with probability at least $1 - \alpha$,

$$\sum_{t=1}^{T}\bar{r}_t \leq \sum_{t=1}^{T}\frac{1}{p_4}\alpha_\rho(x_t) + \sum_{t=1}^{T}(4B + 4\rho G)p_{2,t} + \sqrt{2\ln(1/\delta)\sum_{t=1}^{T}M_t^2}$$

$$\overset{(a)}{\leq}\frac{1}{p_4}\sum_{t=1}^{T}\alpha_\rho(x_t) + C'(4B + 4\rho G) + \frac{(c_f(T) + \rho c_g(T))(4B + 4\rho G)}{p_4}\sqrt{2T\ln(1/\delta)},$$

where (a) we have used the boundedness condition. Note that since $E^{est}$ holds with probability at least $1 - p_1$ for all $t$ and $x$. By a union bound, we have with probability at least $1 - \alpha - p_1$,

$$\sum_{t=1}^{T}d_t \leq \frac{1}{p_4}\sum_{t=1}^{T}\alpha_\rho(x_t) + C'(4B + 4\rho G) + \frac{(c_f(T) + \rho c_g(T))(4B + 4\rho G)}{p_4}\sqrt{2T\ln(1/\delta)}$$

$$= O\left(\frac{1}{p_4}\sum_{t=1}^{T}(c_f(T)\sigma_{t-1}(x_t) + \rho c_g(T)\widetilde{\sigma}_{t-1}(x_t)) + \frac{(c_f(T) + \rho c_g(T))\kappa}{p_4}\sqrt{2T\ln(1/\delta)}\right)$$

$$= O\left(\frac{1}{p_4}c_f(T)\sqrt{T\gamma_T} + \frac{1}{p_4}\rho c_g(T)\sqrt{T\widetilde{\gamma}_T} + \frac{(c_f(T) + \rho c_g(T))\kappa}{p_4}\sqrt{2T\ln(1/\delta)}\right), \quad (22)$$

where $\kappa := 4B + 4\rho G$. Plugging (22) into (19), we obtain that

$$\mathcal{T}_1 + \mathcal{T}_2 \leq O\left(\frac{1}{p_4}c_f(T)\sqrt{T\gamma_T} + \frac{1}{p_4}\rho c_g(T)\sqrt{T\widetilde{\gamma}_T} + \frac{(c_f(T) + \rho c_g(T))\kappa}{p_4}\sqrt{2T\ln(1/\delta)}\right)$$

$$:= \chi(T, \phi).$$

Note that here $\chi(T, \phi)$ is independent of $\phi$ since we have bounded it by $\rho$ in the analysis. Finally, plugging $\chi(T, \phi)$ into (14) and (17) yields the results of Theorem 2. $\square$

## C  Flexible Implementations of RandGP-UCB

In this section, we will give more insights on the choices of $\widehat{\mathcal{D}}$, i.e., sampling distribution for $\widehat{Z}_t$. In particular, we consider the unconstrained case for useful insights with black-box function being $f$.

By the definition of RandGP-UCB, for each $t$, the estimate under RandGP-UCB is given by
$$f_t(x) = \mu_{t-1(x)} + Z_t \sigma_{t-1}(x),$$
where $Z_t \sim \mathcal{D}$. First, by Lemma 3, we have with high probability
$$f(x) \leq \mu_{t-1} + \beta_t \sigma_{t-1}(x),$$
which directly implies that in order to guarantee $E_t^{anti}$ happens with a positive probability, one needs to make sure that $\mathbb{P}(Z_t \geq \beta_t) \geq p_3 > 0$. Thus, one simple choice of $\mathcal{D}$ is a uniform discrete distribution between $[0, 2\beta_t]$ with $N$ points. Then, it can be easily checked that $\mathbb{P}_t\left(E_t^{anti}\right) \geq p_3 > 0$ and also $\mathbb{P}_t\left(E_t^{conc}\right) = 1$ with $c_{f,t}^{(2)} = 2\beta_t$. In addition to uniform discrete distribution, one can also use discrete Gaussian distribution within a range $[L, U]$ as long as $U$, $L$ are properly chosen. Of course, there are many other choices as long as the insight shown above is satisfied, and hence RandGP-UCB provides a lot of flexibility in the algorithm design.

## D  Details on Heavy-Tailed Real-World Data

This dataset is the adjusted closing price of 29 stocks from January 4th, 2016 to April 10th 2019. We use it in the context of identifying the most profitable stock in a given pool of stocks. As verified in [47], the rewards follows from heavy-tailed distribution. We take the empirical mean of stock prices as our objective function $f$ and empirical covariance of the normalized stock prices as our kernel function $k$. The noise is estimated by taking the difference between the raw prices and its empirical mean (i.e., $f$), with $R$ set as the maximum. The constraint is given by $g(\cdot) = -f(\cdot) + h$ with $h = 100$ (i.e., $h \approx B/2$). We perform 50 trials (each with $T = 10,000$) and plot the mean along with the error bars.

## E  More Details on Zero Constraint Violation

**Claim 1.** $T\mathbb{E}_{\pi^*}[f(x)] - T\mathbb{E}_{\pi_\varepsilon^*}[f(x)] \leq \frac{2BT\varepsilon}{\delta}$.

To show this, we let $\pi_\varepsilon(x) := (1 - \frac{\varepsilon}{\delta})\pi^*(x) + \frac{\varepsilon}{\delta}\pi_0(x)$, where $\pi^*$ is the optimal solution to the original baseline problem and $\pi_0$ is the Slater's policy satisfying Slater's condition. First, we note that $\pi_\varepsilon$ is a feasible solution to the new baseline problem introduced above. To see this, we note that $\pi_\varepsilon(x) \geq 0$ and
$$\mathbb{E}_{\pi_\varepsilon}[g(x)] = (1 - \frac{\varepsilon}{\delta})\mathbb{E}_{\pi^*}[g(x)] + \frac{\varepsilon}{\delta}\mathbb{E}_{\pi_0}[g(x)] \leq 0 + (-\varepsilon) = -\varepsilon.$$
Since $\pi_\varepsilon^*$ is the optimal solution while $\pi_\varepsilon$ is a feasible one, we have
$$T\mathbb{E}_{\pi^*}[f(x)] - T\mathbb{E}_{\pi_\varepsilon^*}[f(x)] \leq T\mathbb{E}_{\pi^*}[f(x)] - T\mathbb{E}_{\pi_\varepsilon}[f(x)]$$
$$= T\left(\mathbb{E}_{\pi^*}[f(x)] - (1 - \frac{\varepsilon}{\delta})\mathbb{E}_{\pi^*}[f(x)] - \frac{\varepsilon}{\delta}\mathbb{E}_{\pi_0}[f(x)]\right)$$
$$\leq \frac{2BT\varepsilon}{\delta},$$
where in the last step, we use the boundedness of $f$. Therefore, one can properly choose $\varepsilon$ such that the subtraction of $\varepsilon T$ in the constraint violation can cancel the leading term $O(\sqrt{T})$ (hence bounded or even zero constraint violation) while only incurring an additional additive term of the same order in the regret.

**Remark 4.** *We believe that the same slackness trick can improve the existing $\widetilde{O}(\sqrt{T})$ constraint violation in MDPs [11, 12].*

## F  Discussion on Alternative Method

To the best of our knowledge, there exist two popular methods for analyzing constrained bandits or MDPs. They are both based on primal-dual optimization and differ mainly in the analysis techniques. The first one is based on convex optimization tools as in [11, 12] and our paper. The other one is based on Lyapunov-drift arguments as in [13–15]. For simplicity, we call the first method *convex-opt* method and the second one as *Lyapunov-drift* method. Before we provide further discussion, one thing to note is that all existing works only deal with UCB-type exploration for tabular or linear functions, while our paper is the first one that studies general functions with general exploration strategies beyond UCB.

---
**Algorithm 2** Algorithm in the Lyapunov-drift method
---
1: **Parameters:** $V, \varepsilon, Q(1) = 0$
2: **for** batch $t = 1, 2, \ldots$ **do**
3:     Generate estimate $f_t(x), g_t(x)$ and truncate them to $\bar{f}_t, \bar{g}_t$
4:     Pseudo-acquisition function: $z_t(x) = \bar{f}_t(x) - \frac{1}{V}Q(t)\bar{g}_t(x)$
5:     Choose action $x_t = \arg\max_{x \in \mathcal{X}} z_t(x)$; observe reward $r_t$, and cost $c_t$
6:     Pseudo-acquisition function: $\widehat{z}_{\phi_t}(x) = \bar{f}_t(x) - \phi_t \bar{g}_t(x)$
7:     Update virtual queue: $Q(t+1) = [Q(t) + \bar{g}_t(x_t) + \varepsilon]_+$
8:     Posterior model update: using observations to update model
9: **end for**
---

Now we first briefly explain the main idea behind the Lyapunov-drift method when applied to our setting (for the UCB exploration only). It basically has the same algorithm as the convex-opt method. One minor change is that in Lyapunov-drift method, the dual variable is not truncated by $\rho$ and is denoted by $Q(t)$, since this dual update is similar to a typical queue length update in queueing theory, i.e., truncated at zero; see Algorithm 2. To bound the regret, Lyapunov-drift method decomposes it as the following one, where $\varepsilon \leq \delta/2$ is the slackness as in the last section.

$$\mathcal{R}_+(T) = T\mathbb{E}_{\pi^*}[f(x)] - \sum_{t=1}^T f(x_t)$$

$$= \underbrace{T\mathbb{E}_{\pi^*}[f(x)] - T\mathbb{E}_{\pi^*_\varepsilon}[f(x)]}_{\text{Term 1}} + \underbrace{\sum_{t=1}^T \int_{x \in \mathcal{X}} \left(f(x) - \bar{f}_t(x)\right)\pi^*_\varepsilon(x)\,dx}_{\text{Term 2}}$$

$$+ \underbrace{\sum_{t=1}^T \int_{x \in \mathcal{X}} \bar{f}_t(x)\pi^*_\varepsilon(x)\,dx - \bar{f}_t(x_t)}_{\text{Term 3}} + \underbrace{\sum_{t=1}^T \bar{f}_t(x_t) - f(x_t)}_{\text{Term 4}}. \tag{23}$$

From this, one can see that Term 1, Term 2, and Term 4 can be easily bounded under UCB-type exploration. In particular, by optimism and well-concentration of $\bar{f}_t$, one has Term $2 \leq 0$ and Term $4 = \widetilde{O}(\sqrt{T})$ (we ignore $\gamma_T$ term in this section for simplicity). Moreover, Term 1 enjoys the bound as in Claim 1. Thus, the only challenge is to bound Term 3, which cannot be naturally bounded by greedy selection as in the standard way, since in the constrained case, the greedy selection is with respect to the combined function. To handle this, one needs the following result, which not only helps to bound Term 3, but also is the key in bounding the constraint violation.

**Lemma 4.** *Let* $\Delta(t) := L(Q(t+1)) - L(Q(t)) = \frac{1}{2}(Q(t+1))^2 - \frac{1}{2}(Q(t))^2$. *For any* $\pi$, *we have*

$$\Delta(t) \leq -V\left(\int_{x \in \mathcal{X}} \bar{f}_t(x)\pi(x)\,dx - \bar{f}_t(x_t)\right) + \frac{1}{2}(G + \varepsilon)^2 + Q(t)\left(\int_{x \in \mathcal{X}} \bar{g}_t(x)\pi(x)\,dx + \varepsilon\right). \tag{24}$$

*Proof.* See Appendix F.1. □

Thus, one can see that the first term on the RHS of (24) exists in Term 3 if one chooses $\pi = \pi^*_\varepsilon$. By the optimism $\bar{g}_t(x) \leq g(x)$ and the definition of $\pi^*_\varepsilon$, with a telescope summation, one can easily bound Term 3, hence the regret bound.

**Comparison in regret analysis.** Compared to the regret decomposition in our paper (i.e., (5) and (6)), (23) in the Lyapunov-drift method is more tailored to UCB-type exploration in the sense that the Term 3 is upper bounded separately using the optimism. As a result, it is unclear to us how to generalize it to handle general exploration strategies where one often need to bound Term 2 + Term 3 + Term 4 together and optimism does not hold in general. In contrast, our decomposition (5) and (6) basically keep the same pattern as in the unconstrained case, which enables us to utilize this structure to handle general exploration strategies.

We now turn to the constraint violation. By the virtual queue length update in Algorithm 2, the key step behind the constraint violation bound is to bound $Q(T+1)$. To see this, by the virtual queue length update in Algorithm 2, we have

$$Q(T+1) \geq \sum_{t=1}^{T} \bar{g}_t(x_t) + T\varepsilon = \sum_{t=1}^{T} \bar{g}_t(x_t) - g(x_t) + g(x_t) + T\varepsilon,$$

which implies that

$$\sum_{t=1}^{T} g(x_t) \leq Q(T+1) + \sum_{t=1}^{T} (g(x_t) - \bar{g}_t(x_t)) - T\varepsilon = Q(T+1) + \widetilde{O}(\sqrt{T}) - T\varepsilon,$$

where in the last step we uses the well-concentration of $\bar{g}_t$. To bound the remaining term $Q(T+1)$, the Lyapunov-drift method resorts to a classic tool in queueing theory, i.e., Hajek lemma [16], to bound the virtual queue length at time $T+1$. The idea behind it is simple: if the queue length drift $\Delta(t)$ (defined in Lemma 4) is negative whenever the queue length is large, then $Q(T+1)$ is bounded. To establish the negative drift, one resorts to (24) again by choosing $\pi = \pi_0$. By the definition of $\pi_0$ (Slater's policy), the optimism $\bar{g}_t(x) \leq g(x)$ and boundedness of $\bar{f}_t$, one can easily establish a negative drift, and hence the constraint violation.

**Comparison in constraint violation analysis.** Instead of using Hajek lemma, we directly utilize the convex optimization tool to obtain the constraint violation as in [11, 12], which is conceptually simpler. Moreover, the current constraint violation analysis in the Lyapunov-drift method also relies on the optimism of $\bar{g}_t$, which does not hold in general explorations beyond UCB. On the other hand, one possible limitation of convex-opt method is that the constraint violation depends on the maximum information gain of both $f$ and $g$ (for small $T$) while under Lyapunov-drift method and UCB exploration, the constraint violation only depends on $g$. Finally, when applying Hajek lemma to bound the virtual queue length, there exists a subtlety that makes the standard expected version of Hajek lemma fail due to the correlation of virtual queue length $Q(t)$ and $\bar{g}_t$. We give more details on this subtlety in Appendix G.

**Summary.** Both methods are able to establish sublinear regret and sublinear constraint violation under UCB. Moreover, with the aid of slackness (i.e., $\varepsilon$) in the dual update, both methods can establish bounded or even zero constraint violation. For general exploration strategies beyond UCB, we tend to believe that convex-opt method has advantages over the current analysis in the Lyapunov-drift method, since the latter explicitly relies on optimism in both regret and constraint violation analysis. On the other hand, Lyapunov-drift method can easily handle an anytime slackness, say, $\varepsilon_t = O(1/\sqrt{t})$ rather than $\varepsilon = O(1/\sqrt{T})$.

### F.1 Proof of Lemma 4

*Proof.* Note that by the update rule of the virtual queue in Algorithm 2 and non-expansiveness of projection, we have

$$\Delta(t) \leq Q(t)(\bar{g}_t(x_t) + \varepsilon) + \frac{1}{2} \left( \bar{g}_t(x_t) + \varepsilon \right)^2.$$

Now we will bound the RHS as follows.

$$Q(t)(\bar{g}_t(x_t) + \varepsilon) + \frac{1}{2} \left( \bar{g}_t(x_t) + \varepsilon \right)^2$$

$$\overset{(a)}{\leq} Q(t)(\bar{g}_t(x_t) + \varepsilon) + \frac{1}{2}(G + \varepsilon)^2$$

$$= -V\bar{f}_t(x_t) + Q(t)\bar{g}_t(x_t) + Q(t)\varepsilon + V\bar{f}_t(x_t) + \frac{1}{2}(G + \varepsilon_t)^2$$

$$\overset{(b)}{\leq} -V \int_{x \in \mathcal{X}} \bar{f}_t(x)\pi(x)\,dx + Q(t) \int_{x \in \mathcal{X}} \bar{g}_t(x)\pi(x)\,dx + Q(t)\varepsilon + V\bar{f}_t(x_t) + \frac{1}{2}(G + \varepsilon)^2,$$

where (a) holds by the boundedness of $\bar{g}_t$; (b) holds by the greedy selection in Algorithm 2. Reorganizing the term, yields the required result. $\square$

# G Subtlety in Applying Hajek Lemma to Constraint Violation

As stated before, the key step behind the constraint violation is to establish a negative drift of the virtual queue and then by Hajek lemma, one can show that the virtual queue is bounded in expectation, which in turn can be used to establish a zero constraint violation with a proper choice of slackness variable (i.e., $\varepsilon$) in the virtual queue update. However, the negative drift condition in the standard Hajek lemma (cf. Lemma 11 in [13]) requires a conditional expectation, i.e., condition on all large enough $Q$, the expected drift is negative. Then, if one directly applies the standard Hajek lemma, she would proceed as follows. The goal is to show that $\mathbb{E}\left[\Delta(t) \mid Q(t) = Q\right] \leq -cQ$ for all large $Q$ and $c$ is some positive constant. Recall the bound on $\Delta(t)$ in (24), by the boundedness and let $\pi = \pi_0$, the key is to show that

$$\mathbb{E}\left[\int_{x \in \mathcal{X}} \bar{g}_t(x)\pi_0(x)\,dx + \varepsilon \mid Q(t) = Q\right] \leq -c. \tag{25}$$

To illustrate the idea, we simply suppose that the Slater's condition is satisfied at a single point $x_0$ and $\varepsilon = 0$. To show the above inequality, she may choose the following direction.

$$\mathbb{E}\left[\bar{g}_t(x_0)|Q(t) = Q\right] = \underbrace{\mathbb{E}\left[\bar{g}_t(x_0) - g(x_0)|Q(t) = Q\right]}_{\text{Term (i)}} + \underbrace{\mathbb{E}\left[g(x_0)|Q(t) = Q\right]}_{\text{Term (ii)}} \leq -c.$$

For Term (ii), it is easily bounded by Term (ii) $\leq -\delta$ via Slater's condition since $g(\cdot)$ is a fixed function. To bound Term (i), she may resort to the standard self-normalized inequality for linear bandits and the definition of UCB exploration (cf. [48]). By these standard results, she can show that for any fixed $\alpha \in (0, 1]$, the following holds:

$$\mathbb{P}\{\forall x, \forall t, \bar{g}_t(x) \leq g(x)\} \geq 1 - \alpha. \tag{26}$$

That is, $\bar{g}_t$ is optimistic with respect to $g$. Then, by setting $\alpha = 1/T$ and using the boundedness assumption of both $\bar{g}_t$ and $g$, she may conclude that Term (i) $= O(1/T)$. Unfortunately, the bound on Term (i) is ungrounded since it is obtained by treating the conditional expectation in Term (i) as an unconditional expectation. The subtlety here is that one cannot remove the condition on $Q(t)$ in Term (i), since $\bar{g}_t$ is *not* independent of $Q(t)$ as both of them depend on the randomness before time $t$. Given a particular $Q(t)$, it roughly means that we are taking expectation conditioned on a particular history (i.e., a sample-path). Under this particular history, (26) does not necessarily hold, and moreover, the concentration of $\bar{g}_t$ given $Q(t)$ is hard to compute in this case. As a result, the conditional expectation for Term (i) is hard to compute in general.

**One correct way.** Instead of applying the standard expected version of Hajek lemma, one can consider removing the expectation in Hajek lemma by directly showing that $\bar{g}_t(x_0) \leq -c$ almost surely under the "good event". This is exactly the approach used in [15] (cf. Lemma 5.6). In this way, one can show that with a high probability (i.e., under good event), a negative drift exists and hence the constraint violation bound with high probability.

# H Proof of Corollaries

First, we remark that the first event $E^{est}$ in the probability condition of Assumption 3 can be easily obtained by standard GP concentration result. That is, by [5, Theorem 2], we have $\mathbb{P}(\forall x, t, |f(x) - \mu_{t-1}(x)| \leq \beta_t \sigma_{t-1}(x)) \geq 1 - \alpha_f$ for any $\alpha_f \in (0, 1)$, where $\beta_t = B + R\sqrt{2(\gamma_{t-1} + 1 + \ln(1/\alpha_f))}$ (similar for $g$). Thus, we have $\mathbb{P}\left(E^{est}\right) \geq 1 - p_1$ with $p_1 = \alpha_f + \alpha_g$, $c_{f,t}^{(1)} = \beta_t$, and $c_{g,t}^{(1)} = \widetilde{\beta}_t = B + R\sqrt{2(\widetilde{\gamma}_{t-1} + 1 + \ln(1/\alpha_g))}$. Thus, we only need to check probability condition for the remaining two events and the boundedness condition under different exploration methods.

*Proof of Corollary 1.* UCB: By Definition 1, $f_t(x) = \mu_{t-1}(x) + \beta_t \sigma_{t-1}(x)$ and $g_t(x) = \widetilde{\mu}_{t-1}(x) - \widetilde{\beta}_t \widetilde{\sigma}_{t-1}(x)$. From this, we can directly obtain that $E_t^{conc}$ and $E_t^{anti}$ hold with probability one. Moreover, the boundedness condition naturally holds.

TS: By Definition 1, we have that given the history up to the end of round $t - 1$, $f_t(x) \sim \mathcal{N}(\mu_{t-1}(x), \beta_t^2 \sigma_{t-1}^2(x))$ and $g_t(x) \sim \mathcal{N}(\widetilde{\mu}_{t-1}(x), \widetilde{\beta}_t^2 \widetilde{\sigma}_{t-1}^2(x))$. Thus, for any fixed $x \in \mathcal{X}$, by concentration of Gaussian distribution, we have $\mathbb{P}_t(|f_t(x) - \mu_{t-1}(x)| \leq 2\sigma_{t-1}(x)\beta_t\sqrt{\ln t}) \geq 1 - 1/t^2$, and hence, using the union bound over all $x$, we obtain $\forall x, \mathbb{P}_t\left(E_{t,f}^{conc}(x)\right) \geq 1 - 1/t^2$ with

$c_{f,t}^{(2)} = 2\beta_t\sqrt{\ln(|\mathcal{X}|t)}$. Similarly, we have $\forall x$, $\mathbb{P}_t\left(E_{t,g}^{conc}(x)\right) \geq 1 - 1/t^2$ with $c_{g,t}^{(2)} = 2\widetilde{\beta}_t\sqrt{\ln(|\mathcal{X}|t)}$. Hence, by union bound, we have $\mathbb{P}_t\left(E_t^{conc}\right) \geq 1 - p_{2,t}$ with $p_{2,t} = 2/t^2$. Moreover, when $\pi^*$ concentrates on a single point, by standard anti-concentration result of Gaussian distribution (e.g., Lemma 8 in [5]), we have $\mathbb{P}_t(E_{t,f}^{anti}) \geq p$ with $p := \frac{1}{4e\sqrt{\pi}}$. Similarly, we also have $\mathbb{P}_t(E_{t,g}^{anti}) \geq p$. By independent sampling of $f_t$ and $g_t$, we have $\mathbb{P}_t\left(E_t^{anti}\right) \geq p_3$ with $p_3 = p^2$. The boundedness condition holds due to $\sum_{t=1}^T p_{2,t} \leq 2\sum_{t=1}^T 1/t^2 \leq \pi^2/3 := C'$ and $p_4 = O(p^2)$. $\qquad\square$

*Proof of Corollary 2.* By Definition 2, $f_t(x) = \mu_{t-1}(x) + Z_t\sigma_{t-1}(x)$, where $Z_t \sim \mathcal{N}(0, \beta_t^2)$ and $g_t(x) = \widetilde{\mu}_{t-1}(x) + \widetilde{Z}_t\widetilde{\sigma}_{t-1}(x)$, where $\widetilde{Z}_t \sim \mathcal{N}(0, \widetilde{\beta}_t^2)$. By concentration of Gaussian, we have

$$\mathbb{P}_t(\forall x, |f_t(x) - \mu_{t-1}(x)| \leq 2\sigma_{t-1}(x)\beta_t\sqrt{\ln t}) \geq 1 - 1/t^2,$$

thanks to the "coupled" noise. Hence, we have $\mathbb{P}_t\left(E_{t,f}^{conc}\right) \geq 1 - 1/t^2$ with $c_{f,t}^{(2)} = 2\beta_t\sqrt{\ln t}$. Similarly, we have $\mathbb{P}_t\left(E_{t,g}^{conc}\right) \geq 1 - 1/t^2$ with $c_{g,t}^{(2)} = 2\widetilde{\beta}_t\sqrt{\ln t}$. Thus, by the union bound, we have $\mathbb{P}_t\left(E_t^{conc}\right) \geq 1 - p_{2,t}$ with $p_{2,t} = 2/t^2$. By the anti-concentration of Gaussian, we have $\mathbb{P}_t\left(E_{t,f}^{anti}\right) \geq \mathbb{P}_t(Z_t \geq \beta_t) \geq p$, where $p := \frac{1}{4e\sqrt{\pi}}$. Similarly, we have $\mathbb{P}_t(E_{t,g}^{anti}) \geq \mathbb{P}_t(Z_t \leq -\widetilde{\beta}_t) \geq p$. Since the noise $Z_t$ and $\widetilde{Z}_t$ are independent, we have $\mathbb{P}_t\left(E_t^{anti}\right) \geq p_3$ with $p_3 = p^2$. Then, the boundedness condition holds due to $C' = \pi^2/3$ and $p_4 = O(p^2)$. $\qquad\square$