# OpenReview forum: "On Kernelized Multi-Armed Bandits with Constraints"
_NeurIPS.cc/2022/Conference — NeurIPS 2022 Accept_

### Official Review · Reviewer_HvKs · 2022-07-08

**Rating:** 5
**Confidence:** 4
**Soundness:** 3 good
**Presentation:** 3 good
**Contribution:** 2 fair

**Summary:**

The paper considers the problem of black box optimization under soft constraints where the objective of the learner is to maximize a function f subject to $g \leq 0$, where both $f$ and $g$ are elements in (possibly different) RKHS. The objective is to attain a sublinear regret and a sublinear violation of the constraint. The authors propose a general algorithmic framework for such scenarios that works for a class of exploration strategies satisfying certain conditions. It is shown that GP-UCB and GP-TS satisfy these conditions and the regret and violation bounds are derived for these two policies. Supporting numerical evidence has also been provided.

**Questions:**

1. Line 154-155: Better bounds on $\gamma_t$ are known in the literature [1]. Please update the citations accordingly.

2. Why does the definition of $\mathcal{V}(T)$ have a max operation on the entire sum instead of the summands? From what I understand, each summand should have a max operation on it, just like in the case of so called "stronger cumulative constraint violations" considered in the simulations. In simple words, the "soft" constraint implies that one is allowed to violate the constraint every now and then as long as it is not too often. If the max operation is taken as a whole, then being within the constraints "compensates" for the violations, which should not be the case, or at least is not what is being described in the problem setting. Such a setting is more akin to a reward cost system, where the company is using a resources and the overall cost should not exceed a limit in which using less resources at one time instant allows one to use more in the next time instant. Thus, I am a little concerned by the definition of $\mathcal{V}(T)$ and hence whether the algorithm does solve the "soft" constraint problem. It is promising to see good results in the simulations. However, given the authors present this is a theory-guided study, it is important to have a complete theoretical understanding of the situation.

3. It is claimed that the regret is sublinear and so is violation of constraint. However, this is only true for particular values of $\gamma_T$, or equivalently a subset of kernels. This is a drawback associated with UCB type approaches. There are non-adaptive approaches established in the literature that achieve the optimal regret guarantees for all kernels (See [2], [3]). It would make the results stronger if the authors could use those techniques to strengthen the results. Since one just needs to check if the sufficient conditions mentioned in Assumption 3 are being met, I believe it would actually be rather straightforward to incorporate the changes especially given the authors have already developed a unified framework.

4. In theorem 1, the constraint violation has a term of $\hat{\gamma_T} = \max\{ \gamma_T, \tilde{\gamma}_T\}$. It implies that if one of the kernel is very smooth and the other is quite non-smooth, the constraint violation is always governed by the non-smooth one. Can the authors please explain why that is the case? For example, if the kernel corresponding to $g$ is non-smooth, I can see why one might end up violating the constraint often and hence incur a large constraint violation. However, if the kernel is the corresponding to $g$ is the smoother one, then I would expect the constraint violation to be small and not governed the kernel corresponding to $f$.


[1]  Sattar Vakili, Kia Khezeli, Victor Picheny, "On Information Gain and Regret Bounds in Gaussian Process Bandits", Proceedings of The 24th International Conference on Artificial Intelligence and Statistics, PMLR 130:82-90, 2021

[2] Zihan Li, Jonathan Scarlett, "Gaussian Process Bandit Optimization with Few Batches", Proceedings of The 25th International Conference on Artificial Intelligence and Statistics, PMLR 151:92-107, 2022.

[3] Sudeep Salgia, Sattar Vakili, Qing Zhao, "A Domain-Shrinking based Bayesian Optimization Algorithm with Order-Optimal Regret Performance", Advances in Neural Information Processing Systems, 2021

**Limitations:**

N/A.

**Strengths And Weaknesses:**

Strengths:

I think the analysis techniques used in the work are interesting. Even though the ideas are borrowed from existing work, the work draws interesting parallels and extends the analysis from GP-UCB to GP-TS. I liked the discussion on convex-opt and Lyupanov-opt techniques in the Appendix.

Weaknesses:

Please see the next section for questions.

---

> ### Author Response · Authors · 2022-08-02
> **Response to Reviewer HvKs**
>
> Thanks for taking the time to review our work and provide insightful comments! We are glad to hear that you like our analysis for GP-TS and the discussion on the two methods for constrained bandits.
> We present our detailed response as follows. We would also be happy to provide further clarifications if suitable.
>
> ---
> **Q1: reference on better bound for information gain**
>
> **Response**: Thanks for pointing this out. We have already updated it accordingly in the new version (see line 154).
>
> ---
> **Q2: about constraint violation $\mathcal{V}(T)$**
>
> **Response**:
> - We agree with the reviewer that the current definition of $\mathcal{V}(T)$ allows ''compensation'' across rounds. It is worth noting that this metric is not only used in our paper. Rather, it has been commonly used in many recent works on constrained bandits and RL [11][12][[SGS'20, https://arxiv.org/abs/2002.12435][GZS’22, https://arxiv.org/abs/2206.11889]., which serves as a first important step towards understanding the stronger constraint violation. Moreover, some practical applications indeed allow such ''compensation'' such as energy consumption or stability in networking systems [SGS'20, https://arxiv.org/abs/2002.12435].
> - In fact, it is still an open problem as to whether or not computationally efficient approaches like primal-dual can be used to establish a theoretical guarantee on the stronger definition of a constraint violation, pointed out in [11].
> - We remark that our paper takes the first step to establishing regret and constraint violation bounds for strategies beyond UCB.  Moreover, we provide a generic analysis via a new sufficient condition. Furthermore, we also present the first comprehensive comparison of two widely used analytical tools to tackle constrained bandits and RL. From this perspective, we tend to believe that our paper makes an important attempt to fully understand theoretical performance guarantees for constrained kernelized bandits.
>
> ---
> **Q3: about improved regret bound using new algorithm design frameworks**
>
> **Response**:
> - We agree with the reviewer that there are exciting recent advances in establishing improved regret $\tilde{O}(\sqrt{\gamma_T T})$ rather than $\tilde{O}(\gamma_T \sqrt{T})$, in the **unconstrained** setting though. As suggested by the reviewer, this improved regret is achieved via phase (arm) elimination based on batch pure exploration [LS’22, https://proceedings.mlr.press/v151/li22a.html] or a tree-based domain-shrinking algorithm [SVZ’21, https://arxiv.org/abs/2010.13997]. In addition to these two new algorithmic frameworks, we are also aware of other frameworks that enjoy $\tilde{O}(\sqrt{\gamma_T T})$ regret, such as RIPS [CJS’21, https://arxiv.org/abs/2105.05806] and sparse version of batch exploration [VSSB’22, https://arxiv.org/abs/2202.04005]. All of these are different algorithmic frameworks compared to the standard unconstrained case as in [5,6].
> - In this paper, we mainly focus on incorporating soft constraints into the standard framework. Note that our sufficient condition currently is only established for the CKB algorithm, which is based on the standard unconstrained framework. Whether or not a similar sufficient condition holds for the other aforementioned frameworks is definitely an interesting question, which however is beyond the scope of this paper.
>
> - We would like to highlight the reason why we focus on the standard framework. (i) The existing work on constrained kernelized bandits (i.e., with hard constraints) also builds upon the standard framework [7-10]. Thus, for a fair comparison, we consider the same framework. (ii) The standard framework is not only widely studied by researchers, but is also the default implementation in open-source libraries.  (iii) The standard framework naturally enjoys close relationships with other constrained online learning problems (e.g., linear bandits or linear MDPs). Thus, our results could potentially shed light on other problems, e.g., kernelized MDPs.
>
> That being said, we thank the reviewer for pointing out this great future research direction. We have cited all of the above works and pointed out this interesting direction in the conclusion (see lines 379-381).
>
> ---
>
> **Q4: about the dependence on the information gain of $f$ in constraint violation**
>
> **Response**:
> - We remark that the dependence on the information gain of $f$ in the constraint violation is due to our unified analysis based on the convex-opt method. In fact, if one uses the Lyapunov-drift method, the constraint violation only depends on the information gain of $g$. However, as already mentioned in the paper, the Lyapunov-drift method currently can only handle UCB strategy instead of a class of strategies as we consider. It is also worth noting that our constraint violation can indeed be zero for a large $T$.
> - Thanks for the good point. We have added the above discussion when comparing the convex-opt and Lyapunov-drift methods (see lines 727-729).

---

> > ### Comment · Reviewer_HvKs · 2022-08-04
> > **Acknowledgement of Rebuttal**
> >
> > Thanks for your response.
> >
> > 2. Regarding constraint violation $\mathcal{V}(T)$: There are applications that indeed allow for "compensation" as mentioned by the authors. However, from the motivation section, these applications do not seem to be the main focus of the work. If the model considered in this work allows for "compensation" (which is fine), then I would recommend the authors to be clear and upfront about it. Currently, the paper motivates constraint violation with the stronger condition in mind and suddenly considers the softer version in the definition. I would suggest the authors to update that section and explicitly state the kind of constraint violation considered in this work.
> >
> > 4. Information gain: It is interesting to see that one method yields the information gain of $f$ in the bounds, while the other yields that of $g$. As a follow-up question, is it possible to take a minimum of these to approaches and get a tighter bound on regret?

---

> > > ### Author Response · Authors · 2022-08-06
> > > **Response to follow-up questions**
> > >
> > > Thanks very much for your valuable suggestions and insightful follow-up question. We will answer them in detail as follows and hope they will resolve your concern. We would also be happy to provide further clarifications if suitable.
> > >
> > > > Regarding constraint violation $\mathcal{V}(T)$: There are applications that indeed allow for "compensation" as mentioned by the authors. However, from the motivation section, these applications do not seem to be the main focus of the work. If the model considered in this work allows for "compensation" (which is fine), then I would recommend the authors to be clear and upfront about it. Currently, the paper motivates constraint violation with the stronger condition in mind and suddenly considers the softer version in the definition. I would suggest the authors to update that section and explicitly state the kind of constraint violation considered in this work
> > >
> > > Thank you for your suggestion. We have updated the motivation section by using the following example (highlighted in blue color in the revised version), which is inspired by your initial comment.
> > >
> > > For example, in a wireless networking system, the reward could be the throughput while the constraint is that the average energy consumption is below a threshold. In this case, the constraint can be violated at any time, and moreover, using less energy at one time instant allows one to use more in the next time instant, i.e., ``compensation'' across time (see a formal definition in Eq. (1)).
> > >
> > > > Information gain: It is interesting to see that one method yields the information gain of $f$ in the bounds, while the other yields that of $g$. As a follow-up question, is it possible to take a minimum of these to approaches and get a tighter bound on regret?
> > >
> > > Thanks for this insightful question. Let us first clarify that using the convex-opt method (as in our current main paper), the constraint violation (without introducing the slackness) depends on both $f$ and $g$, while using the Lyapunov-drift (under UCB only), the constraint violation only depends on $g$. Thus, if the question is about taking the minimum of the above two constraint violation bounds, then it is basically the one obtained from Lyapunov-drift (which only depends on $g$) as the other one depends on the maximum of the information gain of $f$ and $g$.
> > >
> > > We also tend to believe that in Theorem 1, the dependence of the constraint violation on the information gain of $g$ is the best one can achieve (when no slackness is introduced) and that no tighter bound on the constraint violation (in terms of the information gain of $f$ and $g$) can be obtained.
> > >
> > >
> > > Finally, we would also give two remarks:
> > >
> > > - In both methods (i.e., convex-opt and Lyapunov-drift), one can introduce the slackness to get zero constraint violation for a large enough $T$. Thus, in this case, the constraint violation does not depend on the smoothness of $f$ or $g$.
> > >
> > > - The dependence on the smoothness of both $f$ and $g$ for the constraint violation in the **convex-opt** method is common, which is due to its unique proof structure based on the primal-dual framework. For example, in both [11,12] for tabular MDPs and linear MDPs, respectively, the constraint violation bound also depends on the
> > > ''smoothness'' of both $f$ and $g$.

---

> > > > ### Comment · Reviewer_HvKs · 2022-08-08
> > > > **Acknowledgement**
> > > >
> > > > Thank you for your detailed response. I don't have any further questions.

---

> > > > > ### Author Response · Authors · 2022-08-09
> > > > > **Thanks for the acknowledgement**
> > > > >
> > > > > We are glad to hear that the reviewer finds our rebuttal satisfactory. We thank the reviewer again for the appreciation of our discussion and comparison of convex-opt and Lyapunovp-drift methods for constrained bandits/RL, which is indeed one of our main contributions.

---

### Official Review · Reviewer_Q2Vc · 2022-07-09

**Rating:** 6
**Confidence:** 3
**Soundness:** 3 good
**Presentation:** 3 good
**Contribution:** 2 fair

**Summary:**

This paper provides a unified framework CKB for kernelized bandits optimization with unknown kernelized constraints based on primal-dual optimization. This framework can employ general exploration strategies (GP-UCB and GP-TS), and achieve sublinear cumulative regret with sublinear cumulative constraint violation. A new exploration strategy RandGP-UCB is also provided. Experiments on synthetic and real-world data are conducted. The paper is generally well-written with detailed proofs provided in the appendices.

**Questions:**

- What are the definitions of $c_f(T)$ and $c_g(T)$ mentioned in Assumption 3?
- Can CKB use the expected improvement or the max variance strategy $x_t=\argmax_x \sigma_{t-1}(x)$? What are the corresponding $h_t$ ?
- What kind of strategies are likely to satisfy the sufficient condition? Can the sufficient condition prevent CKB from employing some exploration strategies?

**Limitations:**

Comparisons between convex optimization methods and Lyapunov-drift methods have been discussed.

**Strengths And Weaknesses:**

Strengths:
- The CKB framework can employ multiple exploration strategies.
- Under the sufficient condition, CKB can attain sublinear cumulative regret and sublinear cumulative constraint violations.

Weaknesses:
- It is frequently mentioned that CKB can use general exploration strategies. Based on my understanding, this algorithm requires a function estimator $f_t$  rather than an acquisition function. Some exploration strategies like maximum variance do not involve an estimator of the unknown function.

Comments:
- For line 154, a better upper bound on maximum information gain for the Matern kernel has been provided in "On information gain and regret bounds in Gaussian process bandits".

---

> ### Author Response · Authors · 2022-08-02
> **Response to Reviewer Q2Vc**
>
> We appreciate your time and thoughtful evaluation of our paper. We recap your comment and present our detailed response as follows. We would also be happy to provide further clarifications if suitable.
>
> ---
> > […] Based on my understanding, this algorithm requires a function estimator $f_t$ rather than an acquisition function. [...] maximum variance do not involve an estimator of the unknown function.
>
> - First, in our main algorithm (Algorithm 1), the required $f_t$ (or $g_t$) is indeed an acquisition function (such as GP-UCB and GP-TS).
> - Second, we clarify that by ''exploration'' strategies, we do not mean pure exploration strategies like max-variance, which is often used to derive the simple regret (sample complexity, see [VBJBS’21, https://arxiv.org/abs/2108.09262]) rather than the cumulative regret we consider, which requires a balance between exploration and exploitation.
>
> ---
> > [...] a better upper bound on maximum information gain for the Matern kernel has been provided [...]
>
> Thanks for pointing this out. We have already updated it accordingly in the new version (see line 154).
>
> ---
> > What are the definitions of $c_f(T)$ and $c_g(T)$  in Assumption 3
>
> These are some functions of $T$. A more concrete statement of the second condition of Assumption 3 is that there exists some function $c_f(T)$ (similarly $c_g(T)$) such that $c_{f,t}^{(1)} + c_{f,t}^{(2)} \le c_f(T)$. Note that this $c_f(T)$ is readily reflected in the final regret bound and it is not required in the algorithm. We have updated the paper accordingly, see line 249.
>
> ---
> > Can CKB use the expected improvement or the max variance strategy?
>
> - First, we tend to believe that the max variance strategy (which is a pure exploration strategy) cannot be directly used in CKB to derive the cumulative regret bound. The argument is that even in the unconstrained case, a direct use of max variance strategy cannot provide a non-trivial cumulative regret bound. On the other hand, if one uses max variance strategy in the phase elimination framework (rather than our standard non-phase-elimination one), one can achieve cumulative regret bound in the unconstrained case, see [LS’22, https://proceedings.mlr.press/v151/li22a.html]. However, whether or not this can be generalized to the constrained case is beyond the scope of our current work since we mainly focus on the standard non-phase-elimination framework.
>
> - Second, it is still unclear to us whether CKB can use expected improvement (EI) while still attaining theoretical guarantees. Note that even in the unconstrained case, the theoretical analysis of kernelized bandits under EI strategy is very different from the one for GP-UCB and GP-TS, see [WF’14, https://arxiv.org/abs/1406.7758]. Thus, one needs more care when analyzing EI for the constrained case.
>
> - Finally, we would also like to highlight that even for the popular Thompson sampling (TS) strategy, there is no prior theoretical analysis under the (soft) constraints. Our paper not only gives the first theoretical guarantees for TS, but covers other strategies, i.e., the class of RandGP-UCB. From this perspective, our work advances the state-of-the-art for constrained kernelized bandits.
>
> ---
> > What kind of strategies are likely to satisfy the sufficient condition? Can the sufficient condition prevent CKB from employing some exploration strategies?
>
> - Let us first give more intuition on our sufficient condition, which in turn helps to understand what kind of strategies are likely to satisfy it. We can first ignore the boundedness condition, which is purely due to technical reasons. For the first probability condition, it basically requires the acquisition function (or exploration strategy) $f_t, g_t$ to satisfy two properties: (i) With high probability, they are close to the true functions $f, g$; (ii) With some positive probability, they are also optimistic with respect to the true functions at the optimal points. Our analysis shows that any strategies that satisfy these two properties enjoy our theoretical guarantees. It is this intuition that not only enables us to establish theoretical guarantees for GP-TS for the first time, but leads us to derive the class of RandGP-UCB strategies, where different distributions can be used.
>
> - Note that the above condition is only a sufficient condition rather than a necessary condition. Hence, it does not prevent CKB from employing other exploration strategies. On the other hand, our sufficient condition is by no means complete. That is, there might exist some other exploration strategies that can offer a theoretical guarantee while not satisfying our sufficient condition. However, a complete characterization is beyond the scope of this work. Instead, our work is mainly focused on exploring the soft constraints beyond the UCB strategy. Meanwhile, we also present the first comprehensive comparison of two commonly used analytical tools for constrained bandits/RL, i.e., convex-opt and Lyapunov-drift method.

---

> > ### Comment · Reviewer_Q2Vc · 2022-08-08
> > **Acknowledgement of Rebuttal**
> >
> > Thanks for your clarifications. I have no more questions.

---

> > > ### Author Response · Authors · 2022-08-09
> > > **Thanks for the acknowledgement**
> > >
> > > We are glad to hear that the reviewer finds our rebuttal satisfactory. We thank the reviewer again for the appreciation of our sufficient condition.

---

### Official Review · Reviewer_x6gL · 2022-07-11

**Rating:** 6
**Confidence:** 3
**Soundness:** 3 good
**Presentation:** 3 good
**Contribution:** 3 good

**Summary:**

The authors study a stochastic bandit problem when the reward function and constraint function lie in a reproducing kernel Hilbert space (RKHS) with a bounded norm. The paper considers soft constraints that may be violated in any round as long as the cumulative violations are small. They solve the restricted maximization problem using  primal-dual optimization and propose a flexible algorithm for various types of exploration including UCB and TS.  They  also provide a unified analysis with sublinear regret and sublinear constraint violation.

Their main contribution seems to suggest a unified solution for constrained kernelized bandits.  The paper succeds in providing a unified regret analysis that accommodates various exploration tools such as UCB and TS by circumventing previous approaches relying on Lyapunov-drift arguement.


**Questions:**

none

**Limitations:**

yes

**Strengths And Weaknesses:**

Strength:

The authors develop a unified framework for kernel bandits with soft constraints using primal-dual optimization and show sublinear reward regret and sublinear total constraint violation when UCB or TS type of exploration is utilized.

To construct a unified algorithm and regret analysis, they identify a novel sufficient condition.

The paper is well written and clear.

---

> ### Author Response · Authors · 2022-08-02
> **Response to Reviewer x6gL**
>
> Thanks for your positive evaluation and appreciation of our unified algorithm and regret analysis. We are glad to hear that you find our derived sufficient condition to be a novel contribution.

---

### Official Review · Reviewer_JSLq · 2022-07-13

**Rating:** 6
**Confidence:** 2
**Soundness:** 1 poor
**Presentation:** 2 fair
**Contribution:** 3 good

**Summary:**

This paper examines kernelized online optimization problems and propose to tackle the problem by some generalization of primal-dual techniques. The techniques developed in this paper looks plausible but I am not an expert in this area so feel a bit unsure about the depth of the techniques and wish the authors can elaborate a bit more on some conceptual questions.



**Questions:**

Please see above.

**Limitations:**

Same as above.

**Strengths And Weaknesses:**

The paper aims to optimize a function in RKHS, subject to a certain soft-constraint (it can be violated in a sublinear manner). Then it argues that by carefully combining Gaussian processes, UBC, and primal-dual algorithms, a non-trivial algorithm can be designed. I feel the way different techniques are integrated is quite reasonable, and this overall sounds like a nice result. Nevertheless I have a few questions:

1. Is there a nice way to interpret the soft constraint formulation. The formulation smells like a "made-up" constraint because of certain limitation of the new techniques developed.
2. How does this work compare to Srinivas, Krause, Kakade, and Seeger 12? Is that the only difference is that a soft constraint is added (and maybe that's the reason some primal dual is needed)? Also, some submodular techniques seemed to be needed to deal with those "information gain" things. It appears that this kind of techniques are not needed in this paper.  Did the author manage to circumvent the submodular property in a different way, or indirectly used this property somewhere.
3. (likely to be my ignorant): does sublinear regret implies that there exists an offline algorithm that can optimize any function in strongly polynomial time (with error parameter $\epsilon$)? That sounds too good to be true (or maybe standard grid-search also can achieve so)?
4. Related to 2 and 3: does the algorithm have a multi-dimensional generalization? I imagine the result would be exponential in the dimension like SKKS12. So my question at the end is that if exponential regret is allowed, how is this better than standard grid search? Comments on both the practical and theoretical aspects would be helpful .

---

> ### Author Response · Authors · 2022-08-02
> **Response to Reviewer JSLq**
>
> Thanks for taking the time to review our work and provide valuable comments. We are glad to hear that you find our work to be a nice result. In the following, we will elaborate on the conceptual questions raised by your review and would be happy to provide further clarifications if suitable.
>
> ---
> **Q1: soft constraint formulation**
>
> **Response**:
> - The cumulative constraint violation formulation for soft constraints in our paper (see line 115) is in fact widely used in many other contexts of online learning. For example, ref. [13] in paper used it for linear bandits (a special case of our kernelized bandits). For RL, this soft-constraint formulation has also been studied in both tabular [11] [SGS'20, https://arxiv.org/abs/2002.12435] and linear MDPs [12][GZS’22, https://arxiv.org/abs/2206.11889]. Furthermore, beyond bandit feedback, this constraint violation metric has also been adopted in online convex optimization with constraints [YNW’17, https://arxiv.org/abs/1708.03741].
>
> - A nice way to interpret this soft constraint is that in many practical applications, one can allow the constraint to be violated at each step as long as the cumulative amount of constraint violations is small, e.g., energy consumption or average queue length in networking systems [SGS’20].
>
> - Another reason why we call this widely-adopted formulation ''soft'' is to distinguish it from the existing works on kernelized bandits that require a hard constraint, i.e., with a high probability the action at each step satisfies the constraint.
>
> - There are no prior works on **kernelized bandits with soft constraints**, even though this constraint formulation has been used in many other contexts.  Hence, the main motivation of this work is to study this soft constraint formulation in kernelized bandits and advance the state-of-the-art by proposing a generic framework for sublinear regret and zero constraint violations.
> ---
>
> **Q2: Compare with [SKKS'12] and submodular property**
>
> **Response**:
> - First, there are two key differences between our work and [SKKS'12]: (i) as the reviewer pointed out, they consider kernelized bandits (KB) **without** constraints, which we consider; (ii) they only deal with one particular acquisition (exploration) strategy (i.e., GP-UCB). In stark contrast, in our paper, we consider KB with constraints under a class of acquisition (exploration) strategies enabled by our sufficient condition, including GP-UCB, GP-TS, and RandGP-UCB. Note that previous work on constrained linear bandits or MDPs also mainly focuses on UCB exploration strategy. To this end, we adopt the primal-dual approach to handle the constraints in a unified way.
> - The submodular property is used only in bounding the information gain. Since both our algorithm and analysis treat this as a black-box term, we can simply plug in the best bound on information gain without worrying about the details. Hence, we do not have to explicitly deal with submodular properties in our analysis.
>
> ---
> **Q34: multi-dimensional case, regret, computations, and grid search**
>
> **Response**:
>
> - If we understand your multi-dimensional generalization correctly, it means that the action domain $\mathcal{X}$ is within $\mathbb{R}^d$, which is already the case considered in our paper. For example, under the SE kernel, the final cumulative regret is $O((\log T)^{d+1}\sqrt{T})$.
>
> - There are two different types of complexities: sample complexity (also called simple regret) (i.e., how many samples are required to return an $\epsilon$-close policy) and computation complexity. Our cumulative regret can be regarded as a stronger notion than sample complexity since cumulative regret also counts the loss at every step while sample complexity only cares about the final policy (action). In fact, a cumulative regret guarantee can be translated to a sample complexity guarantee (not the computation complexity). For example, as shown in [JBC’17, https://arxiv.org/pdf/1706.00090.pdf], for the SE kernel, using the above cumulative regret, one can get a sample complexity guarantee as $\tilde{O}(\frac{1}{\epsilon^2}(\log \frac{1}{\epsilon})^d)$, which is in sharp contrast to sample complexity $O((L/\epsilon)^d)$ for grid search, even under the additional $L$-Lipschitz condition.
>
> - In addition to GP update, the key computation in KB is the optimization of the acquisition function. There have been many efficient algorithms for this inner optimization step. For example, both Meta's BoTorch [https://botorch.org/] and Github BO [https://github.com/fmfn/BayesianOptimization] use the L-BFGS-B algorithm to optimize the acquisition function at each round. For the computation complexity even under a formal regret guarantee, one can resort to discretization, which grows in the order $O(t^{2d})$ [5]. Combining this with the iterative update of matrix inverse, the overall computational complexity is $O(T^{2d+3})$ while still guaranteeing a  regret bound $\tilde{O}(\gamma_T \sqrt{T})$.

---

> > ### Author Response · Authors · 2022-08-08
> > **Follow up**
> >
> > We thank the reviewer again for the valuable comments!
> >
> > Since the discussion period is ending soon, we just wanted to check in and ask if our rebuttal clarified and answered the conceptual questions raised in the review. We would be very happy to engage further if there are additional questions!
> >
> > Also, we wanted to check whether we could provide any additional clarification regarding the merits of the paper that would convince the reviewer to raise the score.

---

### Meta-Review · Area_Chair_EH4o · 2022-09-02

**Recommendation:** Accept
**Confidence:** Certain

**Metareview:**

The paper provides new techniques (algorithmic as well as analytical) to solve black box optimization of smooth functions with constraints. The reviewers are largely in favor of the paper's contributions, and the author responses have helped to clarify several aspects of the presentation and connections to existing work. Therefore, I recommend that the paper be accepted.

**Award:**

No

---

### Decision · Program_Chairs · 2022-09-14

Accept